# Exploring Question Decomposition for Zero-Shot VQA

**Zaid Khan**[1]   **Vijay Kumar BG**[2]   **Samuel Schulter**[2]   **Manmohan Chandraker**[2,3]   **Yun Fu**[2]

[1]Northeastern University    [2]NEC Laboratories America    [3]UC San Diego

{khan.za,yun.fu}@northeastern.edu
{vijay.kumar, samuel, manu}@nec-labs.com

## Abstract

Visual question answering (VQA) has traditionally been treated as a single-step task where each question receives the same amount of effort, unlike natural human question-answering strategies. We explore a question decomposition strategy for VQA to overcome this limitation. We probe the ability of recently developed large vision-language models to use human-written decompositions and produce their own decompositions of visual questions, finding they are capable of learning both tasks from demonstrations alone. However, we show that naive application of model-written decompositions can hurt performance. We introduce a model-driven *selective decomposition* approach for second-guessing predictions and correcting errors, and validate its effectiveness on eight VQA tasks across three domains, showing consistent improvements in accuracy, including improvements of > 20% on medical VQA datasets and boosting the zero-shot performance of BLIP-2 above chance on a VQA reformulation of the challenging Winoground task. Project Site: https://zaidkhan.me/decomposition-0shot-vqa/

## 1   Introduction

On a question-answering test, humans are able to answer some questions in a single step, while other questions require potential deliberation and second-guessing. Visual question answering (VQA) [1–3] has traditionally been treated as a single-step task. Models only get one chance for each question, and each question receives equal amounts of computation. This is incongruent to the natural human approach to such tasks, where simple perceptual questions are quickly answered, while harder reasoning questions are allocated more time and computation.

The emergence of task decomposition techniques for large language models (LLMs) [4] is a potential solution to this incongruency. Task decomposition techniques *prompt* a LLM to break down an initial complex task into simpler subtasks that can each be solved independently. However, VQA has not benefited from advances in task decomposition techniques for two reasons. First, many task decomposition techniques [5, 6] have only been effective in the regime of very large unimodal LLMs with parameters in the 30B+ range, while the LLMs underlying vision-language models are typically much smaller, only recently reaching $\approx$ 13b parameters for publicly available models[7–9]. Second, existing methods for prompting vision-language models (VLMs) during VQA tasks focus on other use cases, such as providing more examples of the input task [10] or more information about the image [11]. Given the recent emergence of multi-billion scale VLMs, our main research question is:

*Can multi-billion scale vision-language models benefit by approaching reasoning-heavy VQA as a two-step rather than a single-step problem using decomposition?*

To this end, we explore a form of task decomposition called *question decomposition* as a strategy for zero-shot visual question answering with large VLMs. Although question decomposition has been explored for specific unimodal QA[12–14], it has not been explored as a strategy for multimodal

37th Conference on Neural Information Processing Systems (NeurIPS 2023).

tasks such as VQA with emerging large VLMs [7–9, 15, 16], and little is known about the in-context learning ability of emerging large VLMs.

First, we probe the in-context learning ability [17–19] of both LMs and VLMs to exploit oracular question decompositions written by humans. We design experiments to understand whether models can learn to use decompositions without explicit training, and whether they are merely exploiting keywords and surface statistics when they use decompositions. Second, we conduct a series of experiments, again using in-context learning, to understand how well models can *produce* decompositions that correct the errors of a fixed VQA model. Last, we propose and study an entirely model-driven closed-loop approach mimicking a simplified form of a classic human second-guessing strategy: second-guess answers based on how confident you are about them. We conduct experiments across three domains (art, natural images, medical), eight datasets, three model families, and model sizes ranging from 80M to 11B parameters. Our contributions can be listed as follows:

1. We experimentally demonstrate that large VLMs based on instruction-tuned LLMs can use decompositions to improve their predictions without any training, and are not merely exploiting changes in word statistics introduced by the decomposition. (Sec. 3)

2. We quantitatively show that generative, instruction-tuned language models are capable of writing effective decompositions zero-shot, without task-specific training. (Sec. 4)

3. We find that applying decomposition naively to every question instance harms performance rather than helps (Fig. 4), and propose *selective decomposition* (Fig. 3), a modular, model-agnostic, training-free strategy that treats VQA as a two-step task. (Sec. 5)

4. We apply selective decomposition to a testbed of 8 datasets and show that it consistently improves performance (Tabs. 3 and 4), with gains of $> 20\%$ on medical VQA datasets[20–22], and boosts the performance of BLIP-2[7] above chance on the Winoground[23] benchmark when formulated as a VQA task. (Sec. 5).

## 2    Background

### 2.1    Problem Setting

In zero-shot VQA, a model $f : v, q \rightarrow a$ is given an image $v$, a question $q$, and outputs an answer, $a$. Unlike traditional VQA, the model $f(\cdot)$ has never seen $v, q, a$ triplets. In practice, such a setting often occurs when $f(\cdot)$ is a foundation model that contains several billion parameters and has undergone large scale pretraining. It is undesirable to retrain such an $f(\cdot)$ on visual question answering pairs specifically, both for reasons of computational convenience and because finetuning can degrade robustness[24]. The most common case is that $f(\cdot)$ is an autoregressive, generative language model that can optionally be conditioned on the visual modality. We restrict ourselves to such models, which approximate $\Pi_{k=1}^{N} p(t_{k+1}|t_{1:k}, v)$, where $v$ is an image and $t_{1:k}$ is a sequence of language tokens. In a zero-shot VQA setting, it is expected that $f(\cdot)$ understands that it has been given a question $q$ and should produce the correct answer $a$ to the question $q$ in the context of the image $v$ by modeling it as $p(a|v, q)$. This setting is common when evaluating very large frozen models, such as in [10, 11], with the exception that in our case, $f(\cdot)$ is a vision-language model rather than a language-only model.

### 2.2    Question Decomposition

Question decomposition is the task of decomposing a complex main question into one or more simpler subquestions that are logically related to the main question, answering those simpler subquestion(s), and then using the answered subquestion(s) to help in composing a final answer to the complex main question. This is a strategy often used by humans for problem solving. For example, consider a human being confronted by a wild animal they have never seen before. To answer the main question "*does this animal pose a threat to me?*" a human might decompose it into subquestions such as "*does the animal have sharp canine teeth?*" and "*does the animal have forward facing eyes typical of a predator?*" Knowing the answer to even one of these subquestions makes answering the main question much easier.

Adopting the terminology of Sec. 2.1, the task of question decomposition consists of *decomposing* a main visual question $v, q$ into one or more subquestions $(s_1, s_2, \ldots)$, answering those subquestions to

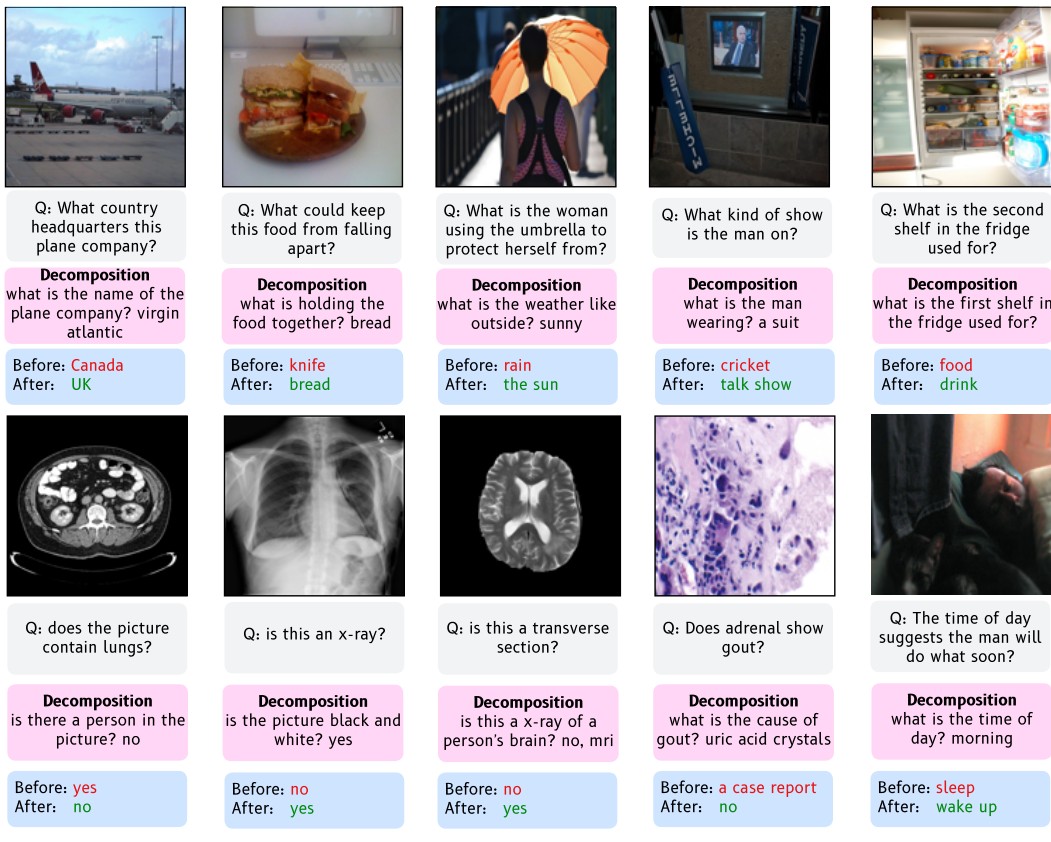

Figure 1: Model-produced decompositions and their error correcting effects. The decompositions and before/after answers shown above were produced by prompting BLIP-2 models based on FLAN-T5 to produce a subquestion, answering the subquestion with the model and feeding the question and answered subquestion back to the model: it is correcting itself. Before answers are wrong and After answers are correct.

obtain the decomposition $((q'_1, a'_1), (q'_2, a'_2) \ldots)$, and then using $v, q$ together with the decomposition $((q'_1, a'_1), (q'_2, a'_2) \ldots)$ to obtain the final answer $a$.

## 2.3 What makes a good subquestion?

In Sec. 2.2, we gave a definition of decompositions that is dependent on notions of "simpler" and "logically related". It is challenging to make these notions precise, and difficult to operationalize them to measure whether a sequence of text really is a valid subquestion according to these notions. To sidestep these difficulties, we adopt a consequentialist view of whether a subquestion is "good", following a common consequentalist tradition in artificial intelligence as a whole [25]. We evaluate the "goodness" of a subquestion by measuring the effect of the subquestion. Concretely, let $v, q, a$ be a visual question triplet where $v$ is the image, $q$ is the question, and $a$ is the answer. Let $p_f(a|v, q)$ be the probability of the ground-truth answer $a$ as assessed by a visual question answering system $f(\cdot)$. We regard a decomposition of $v, q$ consisting of series of subquestions and their answers $((q'_1, a'_1), (q'_2, a'_2) \ldots)$ as "good" if $p_f(a|v, q) < p_f(a|v, q, ((q'_1, a'_1), (q'_2, a'_2) \ldots))$, that is, if seeing the decomposition increases the probability of the ground-truth answer $a$. In practice, we adopt a simpler criterion that takes the consequentialist definition to the limit. *We regard a decomposition as "good" if seeing the decomposition induces the model to produce the true ground-truth answer $a$.*

## 2.4 Scope & Limitations

We only consider in-context learning techniques for zero-shot VQA, and do not explore full model training in this work. The class of model we are interested in are instruction-following vision-language models based on large language models [7–9]. This excludes previous-generation vision-language models that are not based on multi-billion parameter instruction-tuned language models [26–30].

| Decomposition | Image + Text (3B) | | | | Image+Text (13B) | | | |
|---|---|---|---|---|---|---|---|---|
| | Overall | Boolean | Number | Other | Overall | Boolean | Number | Other |
| None (Baseline) | 79 | 82.5 | 6.8 | 67.4 | 79.1 | 81.9 | 13.7 | 70.1 |
| **Oracle/Oracle** | 88.6 | 91.4 | 40.2 | 79.4 | 89.8 | 92.6 | 45.3 | 80.4 |
| Δ w.r.t Baseline | **9.6** | **8.8** | **33.3** | **12.1** | **10.8** | **10.7** | **31.6** | **10.3** |
| **Oracle/Self-Answer** | 84 | 87.3 | 21.4 | 72.8 | 83.9 | 87.1 | 26.5 | 73.2 |
| Δ w.r.t Baseline | 5 | 4.8 | 14.5 | 5.4 | 4.8 | 5.2 | 12.8 | 3.1 |
| **Oracle/No Answer** | 83.3 | 85.9 | 27.4 | 74.9 | 84.1 | 86.9 | 27.4 | 75.2 |
| Δ w.r.t Baseline | 4.4 | 3.4 | 20.5 | 7.6 | 5.1 | 5 | 13.7 | 5.1 |
| **Oracle/Oracle (Scrambled)** | 84.9 | 87.9 | 37.6 | 74.8 | 86 | 88.9 | 39.3 | 76.2 |
| Δ w.r.t Baseline | 5.9 | 5.4 | 30.8 | 7.4 | 6.9 | 7 | 25.6 | 6 |

| Decomposition | Text (3B) | | | | Text (13B) | | | |
|---|---|---|---|---|---|---|---|---|
| | Overall | Boolean | Number | Other | Overall | Boolean | Number | Other |
| None (Baseline) | 57.4 | 64.4 | 6 | 32.2 | 63.8 | 71.9 | 6.8 | 34.3 |
| **Oracle/Oracle** | 72 | 75.8 | 37.6 | 58.4 | 81.5 | 85.1 | 45.3 | 69 |
| Δ w.r.t Baseline | **14.5** | **11.4** | **31.6** | **26.2** | **17.8** | **13.2** | **38.5** | **34.7** |
| **Oracle/Self-Answer** | 62.1 | 65.8 | 23.1 | 48.8 | 68 | 72.1 | 20.5 | 53.7 |
| Δ w.r.t Baseline | 4.6 | 1.4 | 17.1 | 16.7 | 4.3 | 0.2 | 13.7 | 19.4 |
| **Oracle/No Answer** | 64.8 | 68.7 | 21.4 | 50.9 | 75.2 | 79 | 26.5 | 62.2 |
| Δ w.r.t Baseline | 7.3 | 4.3 | 15.4 | 18.8 | 11.4 | 7.1 | 19.7 | 27.9 |
| **Oracle/Oracle (Scrambled)** | 60.5 | 62.6 | 28.2 | 53.3 | 78.9 | 83.1 | 40.2 | 63.5 |
| Δ w.r.t Baseline | 3.1 | -1.8 | 22.2 | 21.1 | 15.1 | 11.3 | 33.3 | 29.2 |

Table 1: Models are capable of using decompositions written by humans to provide more accurate answers. The gray rows are the baseline performance with no decomposition, and each Δ is calculated w.r.t to this baseline. Oracle/Oracle rows denoting oracle subquestions/oracle answers, have the highest Δ. "Self-Answer" means the model answered oracular subquestions itself, and "No Answer" indicates the answer was left out entirely. Image+Text indicates a vision-language model (BLIP-2) was tested with multimodal inputs, while Text indicates the corresponding language model inside BLIP-2 (FLAN-T5) was tested with text only inputs. Validation split of VQA Introspect is the dataset (22k reasoning questions with their associated decompositions).

Not all datasets are suitable for exploring question decomposition, as some primarily test low-level perception skills rather than high-level reasoning skills that would benefit from a decomposition. We thus limit our evaluation to datasets that explicitly test for high-level reasoning / knowledge-based ability. We are few-shot for the task of *visual question decomposition* but zero-shot for the task of *visual question answering*.

# 3   How well can models use decompositions?

Our goal in this section is to understand the ability of vision-language models based on large language models to *consume* decompositions. The hypothesis we test is: ***When provided with gold-standard decompositions on a VQA task, a model's error rate should be lower than without the gold-standard decompositions.*** Evaluating this hypothesis presents a number of challenges. First, how can we obtain a set of decompositions that are apriori "known to be good"? Second, how should the model be fed the decompositions?

To find a source of apriori "good" decompositions, we turn to the literature on internal consistency in visual question answering. To probe consistency in question answering systems, several datasets [31]–[33] have been proposed. A particularly relevant case of such a dataset is VQA-Introspect [32], which probes consistency along a reasoning-perception axis. Selvaraju et al. [32] annotate each question in the VQAv2[1] validation set as a high-level "reasoning" question or a low-level "perception" question. For each "reasoning" question, Selvaraju et al. [32] write 1-3 "perceptual" subquestions which are implied by the reasoning question. For example, given a high-level reasoning question such as "Can I eat this banana?" a model that says "yes" should also reply "yellow" to the low-level perception question "what is the color of the banana?" We propose to use the low-level perception questions and answers written for the high-level reasoning questions as an *oracular* decomposition for the high-level reasoning question, on the basis that the low-level perception questions are simpler than the high-level reasoning question, entail the answer for the high-level reasoning question, and are written by humans.

The second challenge lies in using a decomposition consisting of a series of subquestions and answers $((q'_1, a'_1), (q'_2, a'_2) \ldots)$ alongside a main visual question $(v, q)$. Recall that we cannot train the model $f(\cdot)$ being used for the visual question answering task, and for any arbitrary model, it is unknown whether the model has ever seen the *exact* task of decomposition-aided visual question answering. Thus, we rely on the in-context learning ability [18, 19] of large language models to learn to perform the tasks we require from a demonstration of the task. We handcraft a simple prompt to contain a main visual question $v, q$ from the VQAv2 validation set, along with one human-written oracular subquestion and human-written answer $q', a'$ for the main question $v, q$ extracted from VQA-Introspect. The prompt is simply

```
exemplar = "Context: is the sky blue? no. are there clouds in the sky?
    yes. Question: what weather is likely? Short answer: rain"
prompt = exemplar + "Context: {subquestion}? {subanswer}. Question: {
    question}? Short answer:"
```

**Experiments & Discussion** We use BLIP-2 [7] models based on the instruction-tuned FLAN-T5 [34] in 3B and 13B sizes. Experiments are run on a combination of A6000s and TPUv3s, on the VQA-Introspect validation set containing 22K reasoning questions and their associated decompositions. The results are shown in Tab. 1. Compared to the baseline with no oracular decompositions, both the 3B/13B vision-language models and their corresponding language models show a clear ability to benefit from decompositions across a variety of question types, with numerical questions benefiting the most. Next, we seek to gain insight into the mechanism by which decompositions aid inference. *Is the model merely exploiting changes in surface level statistics?* If so, we would expect that perturbations that leave the statistics largely unchanged but significantly alter the meaning and logical structure of the oracle decomposition should not result in significantly different performance from the unaltered oracle decompositions. We remove the answers from the decomposition so that it only contains the subquestions, and test the effect of only using the subquestions. Compared to the oracle, there is a significant 50% relative decrease in improvement w.r.t to the baseline. Most of the subquestion answers are boolean, so removing them should not significantly change content words in the prompt, though it changes the meaning of the context significantly. Next, we allow the models to answer the subquestions themselves (Oracle/Self-Answer) rather than using the ground-truth questions. The accuracy of all models again decreases relative to the oracle answers, suggesting the answer and question together contribute to the result. Finally, we take the oracle subquestion+answer and scramble the words before providing them to the models. If the model is merely exploiting surface level statistics, the performance difference between the scrambled oracular decompositions and the original decompositions should be minimal, as the words are all the same. Again, we observe a significant drop compared to the original decompositions, suggesting that the models *are not merely exploiting changes in the surface level statistics*. Furthermore, human-written decompositions help in almost all cases over the no-decomposition baseline. **Note:** *See supplement for complete experimental details for all experiments.*

## 4    Can models produce effective decompositions?

In this section, we conduct experiments to answer the following research questions:

1. Can language models $\leq$ 13B parameters learn to produce effective decompositions purely through demonstrations?

2. Is question decomposition mostly a linguistic ability, or is being able to see the image important?

Recall that a decomposition of a visual question $v, q$ is a series of one or more *subquestions* $((q'_1, a'_1), (q'_2, a'_2) \ldots)$ and their answers, with the constraint that the subquestions and answers should have the property that $p_f(a|v, q) < p_f(a|v, q, ((q'_1, a'_1), (q'_2, a'_2) \ldots))$ where $p_f(\cdot)$ represents probability assessed by a given vision-language model $f(\cdot)$ of the ground-truth answer $a$. We simplify this task to the task of producing a *single* subquestion $q'$ given a main visual question $v, q$, and denote the process of decomposition with an arbitrary autoregressive language model $g(\cdot)$ as $d_g(v, q) \rightarrow q'$. We hereafter refer to the model $g(\cdot)$ that generates the decomposition as the *decomposer*. The subquestion is then answered by the vision-language model $f(v, q') = a'$ to produce the subquestion-answer pair $(q', a')$. We call the question answering model the *recomposer*.

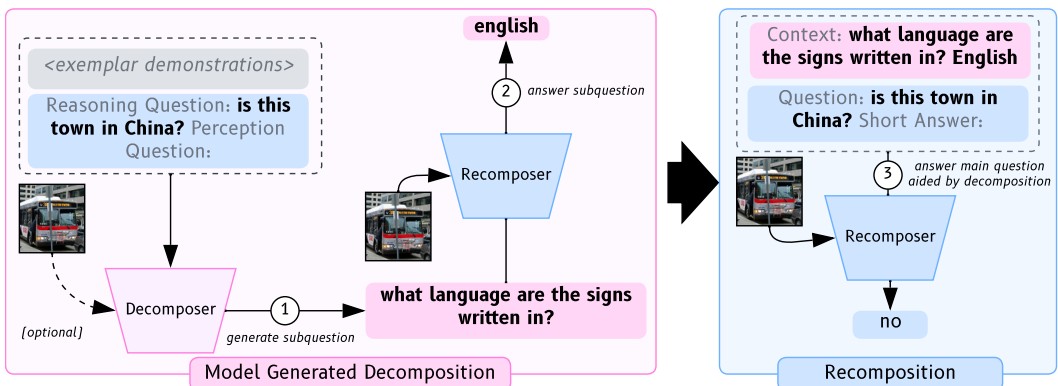

Figure 2: The procedure we use to generate a decomposition and use it as additional guidance during zero-shot VQA. The recomposer can be any question answering model, and the decomposer can be any generative language model, and some models can perform both roles, leading to self-talk. In experiments, we test if various decomposer candidates can learn to write effective subquestions purely from seeing a demonstration of the task.

| VQA Model | Decomposer | A-OKVQA[35] | | | ArtVQA[36] | | | OK-VQA[37] | | | SLAKE[20] | | | Params |
| | | $E_{CR}\uparrow$ | $E_{IC}\downarrow$ | Err | $E_{CR}\uparrow$ | $E_{IC}\downarrow$ | Err | $E_{CR}\uparrow$ | $E_{IC}\downarrow$ | Err | $E_{CR}\uparrow$ | $E_{IC}\downarrow$ | Err | |
|---|---|---|---|---|---|---|---|---|---|---|---|---|---|---|
| | Text | **12.5** | 28.12 | 50.31 | 7.1 | 42.06 | 83.15 | **9.76** | 31.38 | 63.56 | 14.12 | 35.41 | 66.73 | 80.0M |
| | Text | 10.42 | 53.08 | - | 9.56 | 59.81 | - | 9.45 | 52.47 | - | 12.15 | 49.29 | - | 250.0M |
| BLIP2 (3B) | Text | 9.2 | 30.76 | - | **12.22** | 41.12 | - | 8.64 | 29.58 | - | 15.25 | 36.83 | - | 780.0M |
| | Text | 7.99 | 15.11 | - | 6.06 | 21.03 | - | 7.95 | 15.01 | - | 16.38 | 37.68 | - | 3.0B |
| | Image+Text | 7.81 | **10.9** | - | 4.36 | **13.08** | - | 7.42 | **12.29** | - | 15.96 | **28.9** | - | 3.0B |
| | Text | 9.9 | 24.43 | - | 8.05 | 30.37 | - | 9.73 | 22.46 | - | **17.09** | 39.94 | - | 11.0B |
| | Text | 11.52 | 33.44 | 46.99 | 9 | 42.52 | 83.15 | 11.3 | 34.85 | 60.31 | 19.12 | 30.42 | 77.38 | 80.0M |
| | Text | 8.92 | 60.63 | - | 9.94 | 49.07 | - | 9.6 | 58.16 | - | 18.15 | 48.75 | - | 250.0M |
| BLIP2 (11B) | Text | 10.22 | 36.57 | - | **12.12** | 40.65 | - | 11.07 | 35.5 | - | 15.35 | 30.83 | - | 780.0M |
| | Text | 10.78 | **20.59** | - | 8.33 | **19.63** | - | 9.73 | **15.43** | - | 19.85 | 35.42 | - | 3.0B |
| | Image+Text | **14.13** | 26.36 | - | 10.61 | 21.03 | - | 13.54 | 25.06 | - | **20.71** | **30.42** | - | 11.0B |
| | Text | 12.45 | 30.64 | - | 8.05 | 28.97 | - | **12.42** | 27.11 | - | 18.51 | 32.5 | - | 11.0B |

Table 2: Models of drastically different sizes and multimodal capability can produce effective subquestions, as measured by $E_{CR}$ in Eq. (1), their ability to correct errors in a VQA Model. However, subquestions produced by larger models are less likely mislead the consuming VQA model, as measured by $E_{IC}$ in Eq. (2). "Text" indicates a language-only decomposer, while "Image+Text" indicates a vision-language decomposer. "Params" refers to the parameters of the decomposer. A pink highlight indicates when the decomposer and vqa model are the same (the model is talking to itself).

We then measure the effectiveness of the decomposition by measuring the *error correction rate*:

$$E_{CR} = \frac{\sum_{i=1}^{N} \mathbb{1}[f(v_i, q_i) \neq a_i \wedge f(v_i, q_i, (q'_i, a'_i)) = a_i]}{\sum_{i=1}^{N} \mathbb{1}[f(v_i, q_i) \neq a_i]} \quad (1)$$

where $(v_i, q_i, a_i)$ represent the $i$-th image, question, and ground-truth answer respectively, and $q'_i, a'_i$ represent a subquestion generated by the decomposer model and the answer predicted for the subquestion by the recomposer (VQA) model, and $\mathbb{1}[cond]$ is an indicator function that is equal to 1 when $cond$ is true and 0 otherwise. Simply put, $E_{CR}$ measures the number of instances on which $f(\cdot)$ initially predicted a wrong answer, but switched to the correct answer after seeing the decomposition generated by $g(\cdot)$. Alternatively, this can be understood as the effectiveness of a decomposer model at correcting the errors of the recomposer model. The error induction rate $E_{IC}$ is the opposite:

$$E_{IC} = \frac{\sum_{i=1}^{N} \mathbb{1}[f(v_i, q_i) = a_i \wedge f(v_i, q_i, (q'_i, a'_i)) \neq a_i]}{\sum_{i=1}^{N} \mathbb{1}[f(v_i, q_i) = a_i]} \quad (2)$$

and measures how often the produced decompositions flipped an answer that was initially correct to an incorrect answer. The decomposer can be the same as the recomposer if the model can do both tasks by following different prompts, as in the case of instruction-tuned models [38].

**Experiments & Discussion** We use BLIP-2 [7] based on the FLAN-T5[34] as the question answering model (recomposer). For the decomposers, we use FLAN-T5[34] models ranging in size from 80M parameters to 11B parameters, as well as the BLIP-2 models themselves. We use four VQA

datasets from three domains: ArtVQA[36] (art), SLAKE[20] (medical), and A-OKVQA[35] and OKVQA [37] (external knowledge VQA on natural images). We then carry out the procedure illustrated in Fig. 2 for each combination of decomposer, recomposer, and dataset. We handcraft three demonstrations of writing a subquestion for a question, in the form "*Reasoning Question: <question>? Perception Question: <subquestion>?*" For each $v, q$ pair in a dataset, we prompt the decomposer with the demonstration, followed by the question $q$ as in Fig. 2, and measure $E_{CR}$ as in Eq. (1) and $E_{IC}$ as in Eq. (2) for each dataset. We show the results in Tab. 2. We find that *yes, language models $\leq$ 13B parameters can learn to produce effective decompositions just by viewing examples*. Decomposer size correlates positively with $E_{CR}$ ($R^2 = 0.344$), and negatively with $E_{IC}$ ($R^2 = .273$) and the correlations are significant at $\alpha = 0.05$ across a larger collection of eight datasets used in Sec. 5. A human examination of the "subquestions" produced by smaller models shows that many of them are gibberish and not properly formed questions at all. Despite this, they surprisingly manage to maintain an $E_{CR}$ that is sometimes higher than larger models. Finally, **the ability to decompose questions in the evaluated datasets *may* be a primarily linguistic ability** in that it is possible to ask effective subquestions about an image without being able to see the image, and the difference in effectiveness between the Image+Text BLIP-2 models and the text-only FLAN-T5 models of a similar size is on average $\approx 10\%$ of the base error rate (but this may not be true of other VQA datasets).

## 5 Selective Decomposition Works Better Than Naive Decomposition

One problem shows up in Sec. 4, which is that applying decompostions to *every* question can hurt performance, by flipping answers that were initially correct to be incorrect. If we were able to decompose only wrong answers, we would always see a net gain in performance due to the error correction of decompositions. However, in a realistic setting, we do not know apriori that our answers are wrong, and thus run the risk of flipping an answer that was initially correct to be incorrect by applying a decomposition that is misleading. We call this the *second-guessing* problem.

```
v: Image
q: Question
τ: Confidence Threshold

Attempt an initial answer.
â, p(â) = recomposer(v, q)
Selectively decompose q if
answer â is uncertain.
if p(â) <= τ:
    q' = decomposer(v,q)
    a' = recomposer(v, q')
    Answer again.
    â = recomposer(v,q,q',a')
```

Figure 3: Pseudocode for selective decomposition.

To deal with the second-guessing problem, we propose following an intuitive human strategy: stick with your initial answer on questions you are confident about, and only second-guess (apply a decomposition) for questions you are not confident about. Language models can be surprisingly well calibrated[39], meaning that the probability they assess to an output sequence they produce is often well-correlated with the probability that the produced output sequence is the "correct" one for a given task. We make use of this property to treat visual question answering as a selective prediction [40] task, using the language models's confidence as a decision score to determine whether we should apply a decomposition to a instance or stick with the original answer.

We describe the algorithm in pseudocode in Fig. 3. The *selective decomposition* procedure transforms VQA from a single-step task to a two-step task given a decomposer model, a recomposer model, a confidence threshold $\tau$, and a visual question pair $v, q$. An initial answer $\hat{a}$ and confidence $p(\hat{a})$ is solicited from the recomposer model. If $p(\hat{a}) < \tau$, the decomposition procedure is invoked, and a subquestion and answer pair $(q', a')$ are generated by the decomposer and recomposer working together. The recomposer model is then allowed to "second-guess" the inital answer $\hat{a}$ with the decomposition $(q', a')$ as additional context. The decomposer and recomposer can be the same model or different models. We experiment with both scenarios. This introduces an extra hyperparameter $\tau$ into the inference procedure.

**Experiments & Discussion** In Fig. 4, we show the effect of different values of $\tau$ on the accuracy of selective decomposition with several decomposers. Across all datasets and all models, there is a wide range of $\tau$ (expressed as percentiles) for which selective decomposition improves predictive accuracy. At the same time, we clearly demonstrate the *second guessing* problem in Fig. 4. Decomposing *every* question often eventually leads to lower accuracy than decomposing no questions at all, because hallucinations and misleading decompositions can flip an initially correct answer to an incorrect answer.

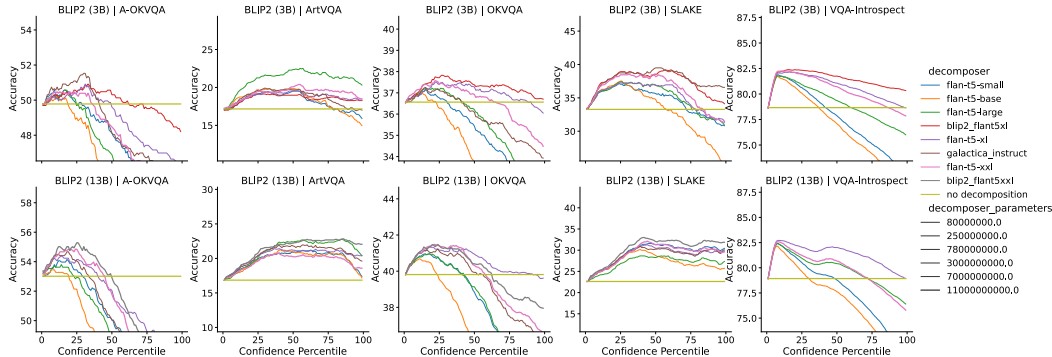

Figure 4: Selective decomposition mitigates the problem of misleading decompositions. We decompose questions based on model confidence in the initial answer, and show how accuracy initial rises past the baseline as the model mostly second guesses wrong answers, and then drops below the baseline with no decompositions (horizontal line) if too many questions initially answered correctly are second-guessed.

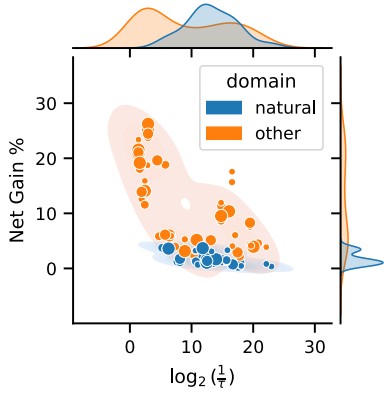

Figure 5: Decomposition is more effective on non-natural image domains, and models are also less confident in these domains. Size of circles is proportional to parameter count.

In Tabs. 3 and 4, we show the highest possible net gain achieved by selective question decomposition on three domains by different decomposers. Selective decomposition consistently improves predictive accuracy regardless of the decomposer and domain. ***Net gains are larger on datasets (passes t-test with $\alpha = 0.05$) containing non-naturalistic images and specialized domains (e.g. medical) than they are on domains containing natural images.*** The mean optimal surprisal $I(\tau)$ for second-guessing answers is lower for non-natural domains ($\mu_{I(\tau)} = 13.2$ for the natural image datasets vs $\mu_{I(\tau)} = 9.0$ for the medical and art datasets, confirmed by t-test at $\alpha = 0.5$). We further visualize this in Fig. 5. *This matches our expectations: you should second guess yourself on domains you understand poorly more than on domains you understand well.* A linear regression fit shows that larger decomposers correlate with larger net gains ($R^2$=0.365, $R^2$=0.342 for natural image domains and medical / art domains respectively, t-test with $\alpha$=0.05).

We reformulate Winoground as a VQA task by turning each caption into a question with a boolean yes / no answer (does "<caption>" describe the image?) on which chance accuracy is 50%. As visible in Tab. 3, all BLIP-2 models perform below random chance, in agreement with previous results on Winoground showing that it is extremely difficult for vision-language models. Surprisingly, after decompositions produced by the relatively FLAN-T5-small/base models (80M/200M parameters), the performance of BLIP-2 (13B) rises to significantly above chance (+18%). Upon inspection, many of the decompositions produced by the model appear to be gibberish, yet remarkably, induce the much larger 13B BLIP-2 model to correct over 30% of its initially wrong answers.

## 6 Literature Review

Task decomposition [5, 6, 41, 42] improves the performance of large language models on zero-shot reasoning tasks. The only work so far to apply similar techniques for VQA is MM-CoT [43], but it does not explore task decomposition with large vision-language models, choosing to finetune a smaller model instead. The ability to use zero-shot task decompositions may be a property of model scale, emerging at 60-200B parameters [44], or may be a property of large-scale pretraining on code [45]. Such large vision-language models have only been developed recently due to advances in vision-language alignment. The prevailing paradigm in vision-language pretraining was to build vision-language models atop (relatively) small language models [26–29, 46] below 1B parameters. Meanwhile, language models were being scaled from 3B-175B parameters [34, 47–50], with each model family having at least one representative with > 10B parameters. Because vision-language

| Decomposer | | AokVQA[35] | | | | OkVQA[37] | | | | VQA Introspect[32] | | | | WinogroundVQA[23] | | | |
|---|---|---|---|---|---|---|---|---|---|---|---|---|---|---|---|---|---|
| VQA | Type | Params | Acc | %↑ | $\eta$ | $I(\tau)$ | Acc | %↑ | $\eta$ | $I(\tau)$ | Acc | %↑ | $\eta$ | $I(\tau)$ | Acc | %↑ | $\eta$ | $I(\tau)$ |
| | T | 80M | | 0.79 | 16 | 13.57 | | 0.59 | 15 | 16.69 | | 3.23 | 8 | 10.64 | | 5.94 | 94 | 0.41 |
| | T | 250M | | 0.44 | 11 | 17.58 | | 0.46 | 13 | 18.05 | | 3.1 | 7 | 13.24 | | **22** | 99 | 0.19 |
| | T | 780M | | 0.87 | 18 | 12.42 | | 0.67 | 26 | 11.02 | | 3.18 | 8 | 10.64 | | 0 | 1 | 8.59 |
| 3B | T | 3B | 49.69 | 1.14 | 31 | 8.13 | 36.54 | 0.97 | 23 | 12.09 | 78.64 | 3.5 | 16 | 5.87 | 45.81 | 0 | 1 | 8.59 |
| | I+T | 3B | | 1.22 | 33 | 7.71 | | **1.29** | 27 | 10.7 | | **3.75** | 20 | 5.24 | | 0 | 1 | 8.59 |
| | T⋆ | 7B | | **1.75** | 31 | 8.13 | | 0.73 | 15 | 16.69 | | 3.37 | 8 | 10.64 | | 0.06 | 2 | 7.4 |
| | T | 11B | | 0.7 | 11 | 17.58 | | 1.03 | 22 | 12.47 | | 3.6 | 14 | 6.27 | | 0 | 1 | 8.59 |
| | T | 80M | | 1.14 | 18 | 14.91 | | 1.15 | 18 | 15.68 | | 3.46 | 8 | 11.81 | | 6.94 | 95 | 0.22 |
| | T | 250M | | 0.35 | 9 | 22.96 | | 0.81 | 11 | 22.13 | | 3.31 | 7 | 15.81 | | **22.12** | 99 | 0.09 |
| 13B | T | 780M | 53.36 | 0.52 | 14 | 17.93 | 39.79 | 1.15 | 18 | 15.68 | 78.93 | 3.46 | 8 | 11.81 | 46.5 | 0.06 | 2 | 3.83 |
| | T | 3B | | 1.05 | 14 | 17.93 | | 1.51 | 18 | 15.68 | | 3.73 | 8 | 11.81 | | 0.69 | 92 | 0.29 |
| | T⋆ | 7B | | 1.48 | 18 | 14.91 | | 1.37 | 24 | 12.54 | | 3.69 | 8 | 11.61 | | 4.06 | 99 | 0.09 |
| | T | 11B | | 1.66 | 25 | 10.95 | | 1.66 | 21 | 13.98 | | 3.68 | 8 | 11.81 | | 0.06 | 14 | 1.64 |
| | I+T | 11B | | **1.92** | 25 | 10.95 | | **1.68** | 24 | 12.54 | | **3.81** | 8 | 11.81 | | 0 | 1 | 5.49 |

Table 3: Increases in accuracy produced by selective decomposition at the optimal second-guessing confidence threshold $\tau$, on external knowledge QA and visual reasoning across several decomposers. $I(\tau) = log_2(\frac{1}{\tau})$ is the surprisal of $\tau$, and $\eta$ is the percent of the questions in the dataset above $I(\tau)$, or equivalently, below $\tau$. T=FLAN-T5, I+T=BLIP-2 (based on FLAN-T5) and T⋆=Galactica.

| Decomposer | | ArtVQA[36] | | | | PathVQA[22] | | | | SLAKE[20] | | | | VQA Rad[21] | | | |
|---|---|---|---|---|---|---|---|---|---|---|---|---|---|---|---|---|---|---|
| VQA | Type | Params | Acc | %↑ | $\eta$ | $I(\tau)$ | Acc | %↑ | $\eta$ | $I(\tau)$ | Acc | %↑ | $\eta$ | $I(\tau)$ | Acc | %↑ | $\eta$ | $I(\tau)$ |
| | T | 80M | | 2.36 | 50 | 10.37 | | **15.89** | 98 | 2.42 | | 3.86 | 24 | 22.12 | | 18.73 | 97 | 2.88 |
| | T | 250M | | 2.68 | 50 | 10.37 | | 11.93 | 69 | 14.79 | | 4.05 | 26 | 16.62 | | 17.57 | 69 | 16.57 |
| | T | 780M | | **5.28** | 55 | 8.91 | | 12.61 | 99 | 1.91 | | 4.15 | 25 | 19.65 | | 15.66 | 69 | 16.57 |
| 3B | T | 3B | 17.17 | 2.44 | 50 | 10.37 | 12.45 | 11.56 | 98 | 2.42 | 33.27 | 3.96 | 38 | 7.33 | 11.7 | 18.82 | 89 | 5.76 |
| | I+T | 3B | | 2.13 | 29 | 17.84 | | 11.34 | 69 | 14.79 | | 5.84 | 61 | 4.71 | | 18.06 | 98 | 1.6 |
| | T⋆ | 7B | | 2.91 | 30 | 17.47 | | 13.9 | 99 | 1.91 | | **6.13** | 50 | 5.71 | | **19.62** | 94 | 4.44 |
| | T | 11B | | 3.15 | 55 | 8.91 | | 14.08 | 98 | 2.42 | | 5.18 | 31 | 10.83 | | 19.17 | 98 | 1.6 |
| | T | 80M | | 4.33 | 63 | 12.09 | | 23.4 | 99 | 1.39 | | 8.95 | 45 | 14.45 | | 23.89 | 99 | 2.95 |
| | T | 250M | | 4.41 | 41 | 19.4 | | 18.46 | 99 | 1.39 | | 7.54 | 39 | 19.48 | | 23.67 | 99 | 2.95 |
| | T | 780M | | 5.91 | 84 | 6.67 | | **22.62** | 99 | 1.39 | | 6.03 | 41 | 17.08 | | 24.02 | 99 | 2.95 |
| 13B | T | 3B | 16.85 | 4.57 | 37 | 20.79 | 5.56 | 21.33 | 99 | 1.39 | 22.62 | 8.67 | 44 | 14.8 | 5.12 | 23.84 | 99 | 2.95 |
| | T⋆ | 7B | | 5.12 | 60 | 13.12 | | 21.07 | 99 | 1.39 | | 8.29 | 39 | 19.48 | | 24.47 | 99 | 2.95 |
| | T | 11B | | 3.94 | 39 | 20.05 | | 21.56 | 99 | 1.39 | | 9.52 | 44 | 14.8 | | **26.25** | 99 | 2.95 |
| | I+T | 11B | | **5.98** | 86 | 6.35 | | 20.85 | 99 | 1.39 | | **10.37** | 42 | 16.08 | | 25.13 | 99 | 2.95 |

Table 4: Increases in accuracy produced by selective decomposition at the optimal second-guessing confidence threshold $\tau$, across two domains (medical/art) and several decomposers. $I(\tau) = log_2(\frac{1}{\tau})$ is the surprisal of $\tau$, and $\eta$ is the percent of the questions in the dataset above $I(\tau)$, or equivalently, below $\tau$. T=FLAN-T5, I+T=BLIP-2 (based on FLAN-T5) and T⋆=Galactica.

pretraining typically requires full model training, aligning these multi-billion parameter models to the visual modality was prohibitively computationally expensive. However, recent discoveries [51, 52] motivated by earlier work with frozen models [53] have shown that the representation spaces of vision models and large-language models are surprisingly close, and rough alignment can be achieved with adapters [15] or linear mapping layers while keeping the language model frozen, and more advanced techniques have given rise to vision-LLMs [7–9]. Our work is closely related to the visual question generation paradigm of [11, 54, 55]. However, we direct our question generation to focus on decompositions rather than general questions.

# 7 Conclusion

We show that question decomposition is already a viable strategy that can be used to implement a more natural approach to VQA. Without any training, instruction-tuned VLMs can learn to produce and use decompositions from a few demonstrations of the task. This approach has many possible future directions. For example, we only consider two-step approaches for visual question answering, where we "hardcode" the depth of the decomposition. A natural next step would be to extend the two-step approach to a multi-step approach, which remains unexplored for large vision-language models in an open-world visual question answering setting. Second, in-context learning has limitations. Would models benefit from being trained to produce and consume decompositions?

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
