The supplementary materials includes a detailed description of implementation details for experiments (Appendix A), a description and statistics of all datasets used (Appendix B), an analysis of the effect of decomposer parameter count on error induction and error correction (Appendix D), and a brief discussion of failure cases (Appendix E).

# A    Experimental Details

## A.1    Models

We use BLIP-2 models built on the FLAN-T5 language model family. We use the official weights and code from LAVIS [56] for the BLIP-2 visual encoder and Q-former. The FLAN-T5 models used in experiments are provided by the Transformers [57] library. The Galactica [47] models we used are instruction-tuned[*] versions of the original galactica models, instruction-tuned on the Evol-Instruct-70k[53] dataset. For all models, we use the official wordpiece tokenizers associated with the model.

## A.2    Image Preprocessing

We use the same image preprocessing as in BLIP-2 [7], which is also identical to the image processing used in [26]. We resize the image to $224 \times 224$ using bicubic interpolation, followed by normalization of pixel values using $\mu = (0.48145466, 0.4578275, 0.40821073)$ and $\sigma = (0.26862954, 0.26130258, 0.27577711)$.

## A.3    Text Preprocessing

We perform no preprocessing of the input text other than padding the batch of input tokens to the length of the largest sequence in the batch. We use the same padding side as the FLAN-T5 models.

## A.4    Inference

We use a batch size of 8 for all datasets and models. We use `bfloat16`[†] precision for FLAN-T5 models (including the FLAN-T5 models inside BLIP-2), and use half-precision (FP16) for the vision encoder inside BLIP-2. The Q-former is kept in full precision. This follows the implementation in [7, 56]. We assign one model per compute device during inference, except when the decomposer and recomposer are the same model, in which case they share the same device.

## A.5    Sampling

### A.5.1    Decomposition

To produce decompositions, we use multinomial beam search sampling with 5 beams and a top-p of 0.95. We use a temperature of 1.0, a length penalty of 1.0, and a reptition penalty of 1.0. These parameters were not optimized, and may be suboptimal.

### A.5.2    Question Answering

We use the same procedure to produce answers for questions with and without decompositions. We use deterministic beam search with 5 beams, restricting the maximum length of the answer to 10 tokens and a minimum of one token. We apply a length penalty of -1.

## A.6    Prompts

### A.6.1    Decomposition

We use the following template to prompt models to produce a decomposition of a reasoning question. The prompt has two exemplars, each consisting of a high-level reasoning question with an associated low-level perceptual subquestion. The exemplars are separated by newlines.

---

[*]https://huggingface.co/GeorgiaTechResearchInstitute/galactica-6.7b-evol-instruct-70k
[†]https://cloud.google.com/tpu/docs/bfloat16

| Dataset | Type | Images | Questions | Avg. Question Length |
|---|---|---|---|---|
| A-OKVQA | external knowledge qa | 1122 | 1145 | 8.70 |
| OK-VQA | external knowledge qa | 5033 | 5046 | 8.09 |
| ArtVQA | fine art vqa | 718 | 1270 | 6.51 |
| VQA-RAD | medical vqa | 314 | 2248 | 6.51 |
| PathVQA | medical vqa | 832 | 6279 | 6.26 |
| SLAKE | medical vqa | 96 | 1061 | 8.11 |
| VQA-Introspect | visual reasoning | 17495 | 22793 | 5.93 |
| Winoground-VQA | visual reasoning | 800 | 1600 | 12.99 |

Table 5: Basic statistics for all eight datasets used in the paper.

```
template = 'Reasoning Question: is the banana ripe enough to eat?
    Perception Question: is the banana yellow?\nReasoning Question: is it
     cold outside? Perception Question: are any people wearing jackets?\
    nReasoning Question: {question} Perception Question:'
```

The galactica-instruct model requires a different prompt, which we describe below. This is because any instructions given to the model have to match the format used in the instruction tuning dataset.

```
template = (
        "Below is an instruction that describes a task. "
        "Write a response that appropriately completes the request.\n
            \n"
        "### Instruction:\n{instruction}\n\n### Response:"
    )
main_question = 'What country is this airline headquartered in?'
prompt = template.format(
        instruction=f"Write a simpler perception question that can
            help to answer: {main_question}"
    )
```

### A.7  Question Answering w/ Decomposition

For question answering without a decomposition, we use the following template:

```
template = 'Question: {question} Short Answer:'
```

This template is identical to that used by [7].

### A.8  Recomposition (Question Answering with Decomposition)

For question answering aided by decomposition, we use the following template (same as the template in Sec. 4). We design the template based on examples from FLAN-T5's training templates [34]. Specifically, we use the keyword `Context:` to identify the start of the decomposition and prepend it to the simple question answering prompt above. Our motivation for the design of this template is that it is conceptually similar to the reading comprehension question answering tasks in FLAN-T5's training data, which demarcate the paragraph to be read using the phrase `Context:`. We expect this similarity to make the task easier for the model.

```
exemplar = "Context: is the sky blue? no. are there clouds in the sky?
    yes. Question: what weather is likely? Short answer: rain"
template = exemplar + "Context: {subquestion}? {subanswer}. Question: {
    question}? Short answer:"
```

## B  Datasets

In Tab. 5, we provide statistics of all datasets used in the paper. We further describe the datasets in this sections.

| VQA Model | dataset
decomposer | aokvqa
$E_{CR}$ ↑ | $E_{IC}$ ↓ | Err | okvqa
$E_{CR}$ ↑ | $E_{IC}$ ↓ | Err | vqa-introspect
$E_{CR}$ ↑ | $E_{IC}$ ↓ | Err | winogroundvqa
$E_{CR}$ ↑ | $E_{IC}$ ↓ | Err | Parameters |
|---|---|---|---|---|---|---|---|---|---|---|---|---|---|---|
| blip2-flant5xl | oracle-decomposer | N/A | N/A | N/A | N/A | N/A | N/A | 51.51 | 8.39 | 22.07 | N/A | N/A | N/A | N/A |
| | flan-t5-small | 12.5 | 28.12 | 50.31 | 9.76 | 31.38 | 63.56 | 39.76 | 19.25 | 22.07 | 42.53 | 38.63 | 54.38 | 80.0M |
| | flan-t5-base | 10.42 | 53.08 | 50.31 | 9.45 | 52.47 | 63.56 | 39.34 | 20.95 | 22.07 | 68.16 | 32.19 | 54.38 | 250.0M |
| | flan-t5-large | 9.2 | 30.76 | 50.31 | 8.64 | 29.58 | 63.56 | 35.49 | 12.68 | 22.07 | 25.17 | 46.3 | 54.38 | 780.0M |
| | blip2-flant5xl | 7.81 | 10.9 | 50.31 | 7.42 | 12.29 | 63.56 | 40.44 | 8.38 | 22.07 | 21.95 | 30.55 | 54.38 | 3.0B |
| | flan-t5-xl | 7.99 | 15.11 | 50.31 | 7.95 | 15.01 | 63.56 | 39.22 | 10.23 | 22.07 | 34.02 | 42.88 | 54.38 | 3.0B |
| | galactica-instruct | 14.76 | 24.25 | 50.31 | 10.48 | 25.23 | 63.56 | 39.46 | 12.46 | 22.07 | 28.16 | 38.22 | 54.38 | 7.0B |
| | flan-t5-xxl | 9.9 | 24.43 | 50.31 | 9.73 | 22.46 | 63.56 | 41.93 | 12.10 | 22.07 | 28.97 | 44.79 | 54.38 | 11.0B |
| blip2-flant5xxl | oracle-decomposer | N/A | N/A | N/A | N/A | N/A | N/A | 58.47 | 10.45 | 21.81 | N/A | N/A | N/A | N/A |
| | flan-t5-small | 11.52 | 33.44 | 46.99 | 11.3 | 34.85 | 60.31 | 43.68 | 21.85 | 21.81 | 43.54 | 36.03 | 53.69 | 80.0M |
| | flan-t5-base | 8.92 | 60.63 | 46.99 | 9.6 | 58.16 | 60.31 | 43.74 | 24.20 | 21.81 | 69.03 | 31.85 | 53.69 | 250.0M |
| | flan-t5-large | 10.22 | 36.57 | 46.99 | 11.07 | 35.5 | 60.31 | 41.77 | 14.12 | 21.81 | 23.75 | 47.64 | 53.69 | 780.0M |
| | flan-t5-xl | 10.78 | 20.59 | 46.99 | 9.73 | 15.43 | 60.31 | 46.06 | 11.94 | 21.81 | 35.74 | 41.7 | 53.69 | 3.0B |
| | galactica-instruct | 12.83 | 33.11 | 46.99 | 13.14 | 29.51 | 60.31 | 46.80 | 15.55 | 21.81 | 33.41 | 30.09 | 53.69 | 7.0B |
| | blip2-flant5xxl | 14.13 | 26.36 | 46.99 | 13.54 | 25.06 | 60.31 | 47.65 | 12.46 | 21.81 | 28.52 | 36.84 | 53.69 | 11.0B |
| | flan-t5-xxl | 12.45 | 30.64 | 46.99 | 12.42 | 27.11 | 60.31 | 46.20 | 16.17 | 21.81 | 28.06 | 42.91 | 53.69 | 11.0B |

Table 6: Error correction and error induction rates for all decomposers on natural image VQA datasets.

| VQA Model | dataset
decomposer | artvqa
$E_{CR}$ ↑ | $E_{IC}$ ↓ | Err | pathvqa
$E_{CR}$ ↑ | $E_{IC}$ ↓ | Err | slake
$E_{CR}$ ↑ | $E_{IC}$ ↓ | Err | vqa-rad
$E_{CR}$ ↑ | $E_{IC}$ ↓ | Err | Parameters |
|---|---|---|---|---|---|---|---|---|---|---|---|---|---|---|
| blip2-flant5xl | oracle-decomposer | N/A | N/A | N/A | N/A | N/A | N/A | N/A | N/A | N/A | N/A | N/A | N/A | N/A |
| | flan-t5-small | 7.1 | 42.06 | 83.15 | 23.54 | 39.64 | 87.55 | 14.12 | 35.41 | 66.73 | 25.64 | 37.26 | 88.3 | 80.0M |
| | flan-t5-base | 9.56 | 59.81 | 83.15 | 18.28 | 44.63 | 87.55 | 12.15 | 49.29 | 66.73 | 24.53 | 44.49 | 88.3 | 250.0M |
| | flan-t5-large | 12.22 | 41.12 | 83.15 | 21.1 | 46.55 | 87.55 | 15.25 | 36.83 | 66.73 | 22.67 | 41.83 | 88.3 | 780.0M |
| | blip2-flant5xl | 4.36 | 13.08 | 83.15 | 16.92 | 37.47 | 87.55 | 15.96 | 28.9 | 66.73 | 23.73 | 24.71 | 88.3 | 3.0B |
| | flan-t5-xl | 6.06 | 21.03 | 83.15 | 19.14 | 42.58 | 87.55 | 16.38 | 37.68 | 66.73 | 24.99 | 30.04 | 88.3 | 3.0B |
| | galactica-instruct | 7.95 | 38.32 | 83.15 | 22.76 | 47.06 | 87.55 | 18.08 | 26.35 | 66.73 | 26.8 | 38.78 | 88.3 | 7.0B |
| | flan-t5-xxl | 8.05 | 30.37 | 83.15 | 22.65 | 46.04 | 87.55 | 17.09 | 39.94 | 66.73 | 25.49 | 33.08 | 88.3 | 11.0B |
| blip2-flant5xxl | oracle-decomposer | N/A | N/A | N/A | N/A | N/A | N/A | N/A | N/A | N/A | N/A | N/A | N/A | N/A |
| | flan-t5-small | 9.0 | 42.52 | 83.15 | 27.49 | 41.83 | 94.44 | 19.12 | 30.42 | 77.38 | 27.0 | 34.78 | 94.88 | 80.0M |
| | flan-t5-base | 9.94 | 49.07 | 83.15 | 21.92 | 40.4 | 94.44 | 18.15 | 48.75 | 77.38 | 26.82 | 26.96 | 94.88 | 250.0M |
| | flan-t5-large | 12.12 | 40.65 | 83.15 | 26.21 | 35.82 | 94.44 | 15.35 | 30.83 | 77.38 | 27.29 | 33.91 | 94.88 | 780.0M |
| | flan-t5-xl | 8.33 | 19.63 | 83.15 | 24.99 | 39.54 | 94.44 | 19.85 | 35.42 | 77.38 | 26.63 | 21.74 | 94.88 | 3.0B |
| | galactica-instruct | 10.42 | 36.45 | 83.15 | 25.3 | 42.98 | 94.44 | 21.92 | 42.5 | 77.38 | 28.18 | 40.87 | 94.88 | 7.0B |
| | blip2-flant5xxl | 10.61 | 21.03 | 83.15 | 24.33 | 34.96 | 94.44 | 20.71 | 30.42 | 77.38 | 28.04 | 24.35 | 94.88 | 11.0B |
| | flan-t5-xxl | 8.05 | 28.97 | 83.15 | 25.28 | 38.97 | 94.44 | 18.51 | 32.5 | 77.38 | 29.35 | 29.57 | 94.88 | 11.0B |

Table 7: Error correction and error induction rates for all decomposers on non-natural image domains (medical and fine art VQA).

**Natural Image Datasets** These include A-OKVQA[35], OK-VQA[37], VQA-Introspect[32], and Winoground[23]. These datasets include natural images only. For A-OKVQA, OK-VQA, and VQA-Introspect, the source of these images is the COCO[59] dataset. While Winoground and VQA-Introspect contain mostly *visual reasoning* that do not require significant external knowledge (e.g. historical facts), OK-VQA and A-OKVQA ask questions which require "outside" factual knowledge to answer, such as historical facts and contemporary information (e.g. which country does a specific airline operate in?).

**Other Domains** Besides natural images, we also use datasets consisting of fine art images [36] and medical images. The datasets consisting of medical images are themselves each drawn from different subdomains of medicine. PathVQA [22] contains pathology images, VQA-RAD [21] contains radiology images, and SLAKE [20] contains general medical images.

## C $E_{CR}$ and $E_{IC}$ for all datasets

In Tabs. 6 and 7, we show $E_{CR}$ and $E_{IC}$ for all decomposers and all datasets used. We note that the oracular decompositions appear to have a similar error induction rate $E_{IC}$ as the best model-generated decompositions (BLIP2-FLANT5XL / BLIP2-FLANT5XXL), but have a noticeably higher error correction rate $E_{CR}$ of +10% relative to the best model generated decompositions. An observation from this is that the model has a limited capacity to reason from decompositions, because even human-generated, oracular decompositions mislead it roughly 8% of the time. Another point of note is that the instruction-tuned Galactica [47] model is not significantly better at writing decompositions than the FLAN-T5 models on medical datasets, despite being trained on much more scientific data.

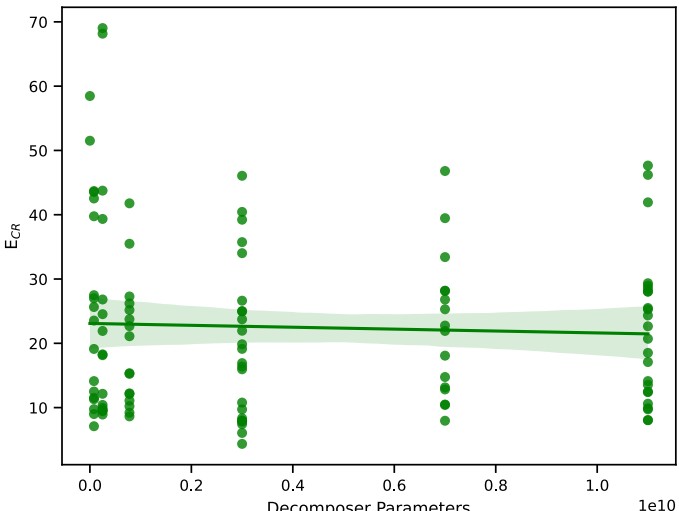

Figure 6: Error correction rate $E_{CR}$ on all datasets (except Winoground) with respect to the number of parameters in the decomposer. There is statiscally significant correlation between the number of parameters $R^2 = 0.40$. The slope is 0.0215 when the unit scale is set to 100M parameters, corresponding to a $\approx .2\%$ increase in $E_{CR}$ for every 1B increase in parameters.

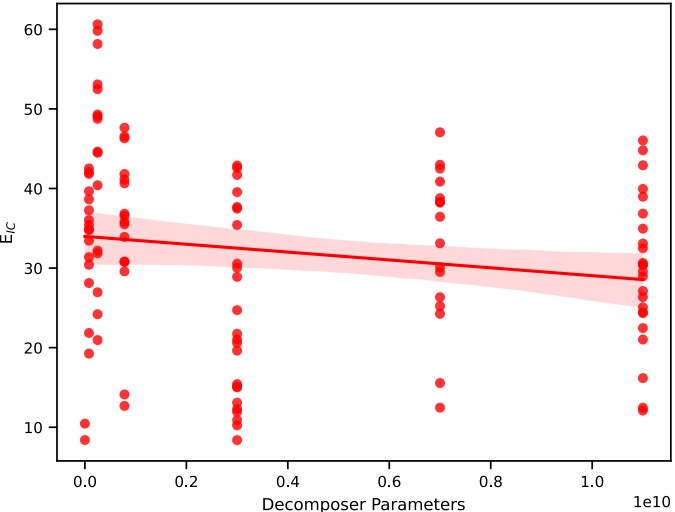

Figure 7: Error induction rate $E_{IC}$ on all datasets (except Winoground) with respect to the number of parameters in the decomposer. There is statiscally significant correlation between the number of parameters $R^2 = 0.35$. The slope is $-0.07$ when the unit scale is set to 100M parameters, corresponding to a $\approx .7\%$ increase in $E_{CR}$ for every 1B increase in parameters.

```
question='what sit around a table?'

Decomposition(
    subquestion='people sit around a table',
    subanswer='in a painting'
)
```

```
question='Does the picture contain colon?'

Decomposition(
        subquestion='Yes, the picture contains a colon. No, the
picture does not contain a colon',
        subanswer='No, the picture does not contain a colon'
    )
```

```
question='What kind of day is it outside?'

Decomposition(
        subquestion='It is a rainy day.',
        subanswer='A dog is sitting on top of a car'
    )
```

Figure 8: Examples of failure cases when attempting to produce decompositions.

## D  $E_{IC}$ Drops Faster Than $E_{CR}$ Rises

In Figs. 6 and 7, we plot the relationship between $E_{CR}$, $E_{IC}$, and parameter count of the decomposer. We exclude Winoground from the plots because the mechanism of effect of decompositions appears to be different for Winoground. There are statistically significant relationships (at the 95% significance level, $\alpha = 0.05$) for both $E_{CR}$ and $E_{IC}$. $E_{IC}$ drops .7% for every 1B increase in parameters, while $E_{CR}$ increases .2% for every 1B increase in parameters. This indicates that the strongest effect of scaling is to produce less misleading decompositions. The ability to produce decompositions that correct more and more errors appears to increase more slowly with scale.

## E  Failure Cases

In Fig. 8, we show examples of failure cases that occur when attempting to produce decompositions. The incidence of failure cases varies by domain and model size. On natural image domains and for large models (3B or more), the number of failure cases is very low. For non-natural image domains (e.g. art), even the largest models have a high incidence of failure cases in which the produced decomposition is not even a question. In some cases (e.g. Winoground) the failed decompositions can still correct the answer, even when they appear to be unrelated to the content of the image. We hypothesize that there is a connection between this failure mode (apparently unrelated text results in the right answer) and the phenomenon of nonsense prompts in discrete prompt tuning [60], in which prefixing an apparently random sequence of words to a prompt results in significantly increased performance.

| | Decomposer | | AokVQA[35] | | | | OkVQA[37] | | | | VQA Introspect[32] | | | | WinogroundVQA[23] | | | |
|---|---|---|---|---|---|---|---|---|---|---|---|---|---|---|---|---|---|---|
| VQA | Type | Params | Acc | % ↑ | η | I(τ) | Acc | % ↑ | η | I(τ) | Acc | % ↑ | η | I(τ) | Acc | % ↑ | η | I(τ) |
| OpenFlamingo | T | 80M | | 9.69 | 99 | 0 | | 9.81 | 99 | 0 | | 21 | 99 | 0.01 | | 26.69 | 99 | 0.02 |
| | T | 3B | | 15.28 | 99 | 0 | | 13.79 | 99 | 0 | | 35 | 99 | 0.01 | | 19.56 | 99 | 0.02 |
| | T | 11B | 0.96 | 12.49 | 99 | 0 | 0.59 | 11.41 | 99 | 0 | 5.68 | 38.23 | 99 | 0.01 | 2.38 | 22.62 | 99 | 0.02 |
| | Galactica | 7B | | 9.78 | 97 | 0.01 | | 11.24 | 99 | 0 | | 35.73 | 99 | 0.01 | | 16.94 | 99 | 0.02 |
| | Falcon | 7B | | 9.52 | 99 | 0 | | 9.47 | 99 | 0 | | 31.41 | 99 | 0.01 | | 28.31 | 99 | 0.02 |
| InstructBLIP | T | 80M | | 20.96 | 99 | 0.03 | | 4.6 | 99 | 0.03 | | 7.68 | 47 | 0.12 | | 18.69 | 99 | 0.05 |
| | T | 3B | | 21.22 | 99 | 0.03 | | 7.59 | 99 | 0.03 | | 6.68 | 48 | 0.12 | | 20.44 | 99 | 0.05 |
| | T | 11B | 24.63 | 20.79 | 99 | 0.03 | 36.78 | 5.87 | 99 | 0.03 | 75.86 | 6.95 | 45 | 0.13 | 12.88 | 16.25 | 99 | 0.05 |
| | Galactica | 7B | | 22.1 | 99 | 0.03 | | 6.68 | 99 | 0.03 | | 7.95 | 52 | 0.11 | | 17.31 | 99 | 0.05 |
| | Falcon | 7B | | 16.51 | 99 | 0.03 | | 2.81 | 99 | 0.03 | | 4.32 | 21 | 0.23 | | 13.13 | 99 | 0.05 |
| BLIP2 (3B) | Galactica | 7B | 49.78 | 1.75 | 31 | 8.13 | 36.52 | 0.73 | 15 | 16.69 | 78.63 | 3.37 | 8 | 10.64 | 45.69 | 0.06 | 2 | 7.4 |
| | Falcon | 7B | | 0.7 | 18 | 12.91 | | 0.79 | 16 | 15.87 | | 3.35 | 8 | 10.61 | | 1.31 | 99 | 0.19 |
| BLIP2 (11B) | Galactica | 7B | 53.19 | 1.48 | 18 | 14.91 | 39.87 | 1.37 | 24 | 12.54 | 78.96 | 3.69 | 8 | 11.81 | 46.38 | 4.06 | 99 | 0.09 |
| | Falcon | 7B | | 1.22 | 13 | 18.63 | | 1.49 | 17 | 16.3 | | 3.58 | 8 | 11.82 | | 7.75 | 99 | 0.09 |

Table 8: Experiments with OpenFlamingo [61], InstructBLIP [62] and Falcon [63] on natural image domains.

| | Decomposer | | ArtVQA[36] | | | | PathVQA[22] | | | | SLAKE[20] | | | | VQA Rad[21] | | | |
|---|---|---|---|---|---|---|---|---|---|---|---|---|---|---|---|---|---|---|
| VQA | Type | Params | Acc | % ↑ | η | I(τ) | Acc | % ↑ | η | I(τ) | Acc | % ↑ | η | I(τ) | Acc | % ↑ | η | I(τ) |
| OpenFlamingo | T | 80M | | 3.39 | 99 | 0 | | 7.26 | 99 | 0.01 | | 17.34 | 99 | 0.01 | | 13.61 | 89 | 0.02 |
| | T | 3B | | 5.83 | 99 | 0 | | 15.81 | 99 | 0.01 | | 27.62 | 99 | 0.01 | | 20.11 | 98 | 0.01 |
| | T | 11B | 0 | 5.04 | 99 | 0 | 3.44 | 16.31 | 99 | 0.01 | 0.85 | 25.82 | 99 | 0.01 | 6.01 | 20.64 | 98 | 0.01 |
| | Galactica | 7B | | 3.07 | 99 | 0 | | 19.05 | 99 | 0.01 | | 22.9 | 99 | 0.01 | | 17.35 | 99 | 0.01 |
| | Falcon | 5B | | 7.8 | 99 | 0 | | | N/A | | | 17.06 | 99 | 0.01 | | 20.33 | 89 | 0.02 |
| InstructBLIP | T | 80M | | 0 | 1 | 0.9 | | 2.05 | 97 | 0.08 | | 4.62 | 43 | 0.29 | | 0.89 | 52 | 0.23 |
| | T | 3B | | 0 | 1 | 0.9 | | 1.64 | 87 | 0.15 | | 4.15 | 92 | 0.1 | | 1.65 | 60 | 0.2 |
| | T | 11B | 32.36 | 0 | 1 | 0.9 | 28.06 | 2.85 | 97 | 0.08 | 28.93 | 4.24 | 43 | 0.29 | 32.61 | 0.8 | 52 | 0.23 |
| | Galactica | 7B | | 0 | 1 | 0.9 | | 2.01 | 86 | 0.15 | | 4.52 | 56 | 0.23 | | 1.33 | 51 | 0.23 |
| | Falcon | 5B | | 0 | 1 | 0.9 | | 1.24 | 76 | 0.19 | | 3.2 | 44 | 0.28 | | 0.85 | 50 | 0.23 |
| BLIP2 (3B) | Galactica | 7B | 17.01 | 2.91 | 30 | 17.47 | 12.45 | 13.9 | 99 | 1.91 | 33.36 | 6.13 | 50 | 5.71 | 11.7 | 19.62 | 94 | 4.44 |
| | Falcon | 5B | | 1.73 | 32 | 16.64 | | 13.71 | 89 | 6.12 | | 5.75 | 26 | 16.67 | | 17.22 | 71 | 13.67 |
| BLIP2 (11B) | Galactica | 7B | 16.85 | 5.12 | 60 | 13.12 | 5.56 | 21.07 | 99 | 1.39 | 22.62 | 8.29 | 39 | 19.48 | 5.12 | 24.47 | 99 | 2.95 |
| | Falcon | 5B | | 3.39 | 41 | 19.39 | | 20.61 | 99 | 1.39 | | 9.52 | 88 | 2.39 | | 23.44 | 99 | 2.91 |

Table 9: Experiments with OpenFlamingo [61], InstructBLIP [62] and Falcon [63] on non-natural image domains.

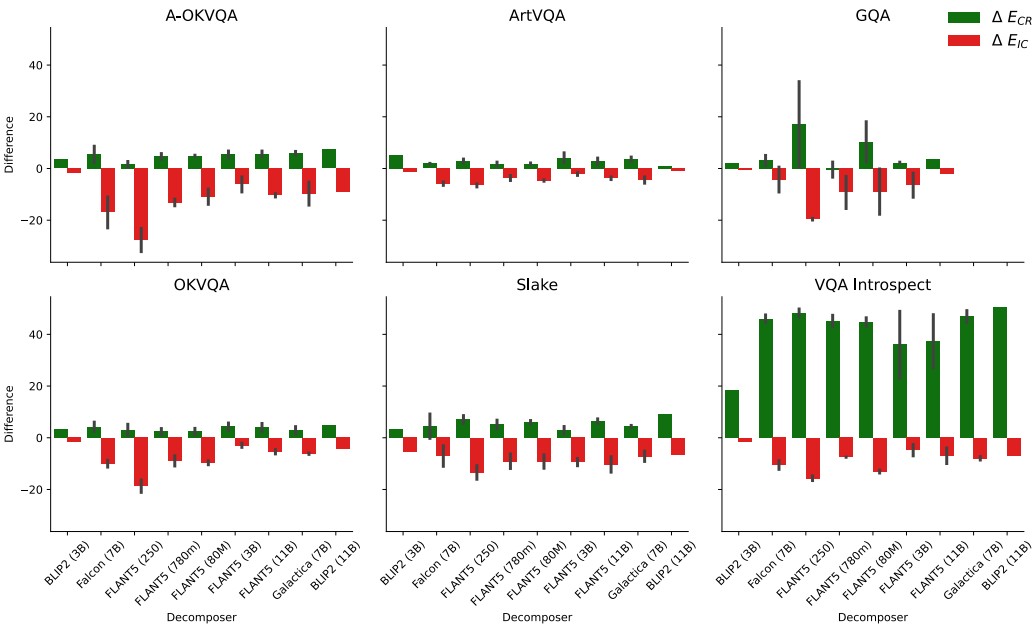

Figure 9: The change in $E_{IC}$ and $E_{CR}$ after selectively decomposing questions.