# OpenReview forum: "Exploring Question Decomposition for Zero-Shot VQA"
_NeurIPS.cc/2023/Conference — NeurIPS 2023 poster_

### Official Review · Reviewer_TpMW · 2023-06-13

**Soundness:** 3 good
**Presentation:** 3 good
**Contribution:** 3 good
**Rating:** 5
**Confidence:** 4

**Summary:**

This paper studies the question decomposition of VQA.
To begin with, the authors demonstrate that large VLMs equipped with LLMs can effectively leverage question decomposition to improve answering performance.
Thereafter, some experiments are conducted to prove that instruction-tuned language models are capable of auto-writing decompositions.
Since some questions do not require decomposition to reach the right answer, the authors then propose a novel selective decomposition method to selectively decide which questions demand decomposition.

**Strengths:**

- The paper is well-written and well-organized. Most parts of this paper are easy to follow.
- The studied problem is interesting and novel. It helps us to understand the in-context learning capability in large VLMs with instruction-tuned LLMs.
- To validate the effectiveness of question decomposition, the authors perform comprehensive experiments.

**Weaknesses:**

- My biggest concern lies in Sec. 4 about whether large models can produce effective decompositions. As shown in Table 2, most $E_{CR}$ values are pretty low while $E_{IC}$ are relatively high. According to the definitions of these two terms, it seems that most decompositions severely hurt the model performance. For example, 10% $E_{CR}$ means only 1 of 10 vqa instances can be corrected by the recomposer. In contrast, the average value of $E_{IC}$ is around 40. This means 4 of 10 vqa instances after decomposition failed.
- For Sec. 5, have authors tried these newly introduced metrics after the selective decomposition?
- Have the authors considered using the GQA dataset [1], which involves a longer chain of reasoning?
- About Table 1, the oracle/no answer outperforms oracle/self-answer by a large performance margin. This result does not make much sense since using more context usually leads to more accurate answer predictions.
- Moreover, the scrambling is not well-explained. How many words are perturbed? This question is important because a larger perturbation often causes a severe performance drop. In addition, why the authors have not tried scrambling the questions? This is also essential because the well-known bias problem in VQA is from question key words and answers.

Typo:
- Fig. 1 is not referred in the main manuscript.

[1] GQA: A New Dataset for Real-World Visual Reasoning and Compositional Question Answering. In CVPR 2019.

**Questions:**

Overall, I believe this is a good paper.
However, the effectiveness of question decomposition from large VLMs is confusing.
If the authors can address this concern, I will raise my initial score.

---

> ### Author Rebuttal · Authors · 2023-08-10
>
> Great questions, and thank you for the positive appraisal of our work. We have added an experiment for you, as well as a detailed discussion of the of $E_{IC}$ and $E_{CR}$ which should clear up any confusion: we do believe large models can decompose well.
>
> ### Why does selective decomposition work in spite of seemingly high error rates?
> There is a subtle error in your interpretation. The $E_{CR}$ and $E_{IC}$ values have to be interpreted with care, because they are relative to the model's error rate. For example, consider a situation in which there are 100 questions, and the error rate is 90%, so the model only predicts $\frac{10}{100}$ correct answers. Assume an $E_{IC}=0.5$ and $E_{CR}=0.1$. This would mean that 5 initially correct answers were flipped to incorrect, but 9 initially wrong answers were flipped to correct answers. So the number of total correct answers would go from $10 \rightarrow 14$, a $40\%$ increase in accuracy.
>
> If we calculated the $E_{IC}$ and $E_{CR}$ absolutely (by making the denominator in Eq 1 & 2) the total number of questions, your interpretation would be correct. However, if we calculate it this way , the $E_{CR}$ and $E_{IC}$ values are very close, which we have computed for you below in the case of BLIP-2 (3B). So in this case, we can see that the $E_{IC}$ and $E_{CR}$ are very close and a small nudge from selective decomposition can increase the $E_{CR}$ over the $E_{IC}$ if it is not already higher.
>
> | Dataset | $E_{CR}$ | $E_{IC}$ | Error Rate |
> |---|---|---|---|
> | A-OKVQA | 3.93 | 5.41 | 50.31 |
> | ArtVQA | 3.62 | 2.2 | 83.15 |
> | OK-VQA | 4.72 | 4.48 | 63.56 |
> | Slake | 10.65 | 9.61 | 66.73 |
> | VQA Introspect | 8.92 | 6.53 | 22.07 |
>
> As a corrollary, consider the case in which most of the model's low confidence answers are wrong, which will happen if the model is well-calibrated . In this case, there is *no harm to second guessing low-confidence questions if your error correction rate is higher than the error rate on low confidence questions!* Because the error rate on low-confidence questions is very high, most of them are already wrong and there is no penalty induced for changing an answer from wrong $\rightarrow$ wrong.
>
> ### $E_{IC}$ and $E_{CR}$ do increase after selective decomposition
>
> We have computed these metrics after decomposition (using the absolute formulation so it is easier to interpret), which can be viewed in the _rebuttal PDF_ (Figure 9). In all cases, selective decomposition results an in increased $E_{CR}$ of the decomposed question is higher, and lower $E_{IC}$.
>
> ### We add experiments on GQA.
>
>
> | VQA Model | Type | Decomposer Params | Acc | $\%\uparrow$ | $\eta$ | $I(\tau)$ |
> |---|---|---|---|---|---|---|
> | 3B | I+T | 3B | 53.79 | 1.52 | 73 | 1.14 |
> |  | T | 80M |  | 0.2 | 3 | 11.4 |
> |  | T | 3B |  | 0.1 | 3 | 11.4 |
> |  | T | 11B |  | 0.81 | 73 | 1.14 |
> |  | Galactica | 7B |  | 0.71 | 8 | 6.51 |
> |  | Falcon | 7B |  | 1.31 | 91 | 0.66 |
> | 13B | I+T | 11B | 55.11 | 0.81 | 49 | 1.95 |
> |  | T | 80M |  | 0.1 | 2 | 16.39 |
> |  | T | 3B |  | 1.42 | 49 | 1.95 |
> |  | T | 11B |  | 0.2 | 2 | 16.39 |
> |  | Galactica | 7B |  | 1.52 | 69 | 1.34 |
> |  | Falcon | 7B |  | 0.4 | 12 | 5.98 |
>
> ### Does self-answering hurt performance?
>
> This is only the case for the text-only models. As they cannot see the image, their self-answers to the decomposition are hallucinated and can mislead themselves. For the image+text models, there is only a substantial difference between self-answer and no-answer in one case: BLIP-2 (3B) on "numerical" questions. The reason for this is simple: upon manually inspecting the answers, we find that the VQA models make substantial errors when they answer a decomposition. This has also been observed by the line of work building upon VQA-Introspect, which has shown that even VQA models with high accuracy are often inconsistent, providing wrong answers to questions that should be entailed by a main question.
>
> This effect can be seen clearly when comparing the oracle/oracle rows with the oracle/self-answer rows -- even self-answering with a strong VQA model reduces the benefit of the decomposition, showing that the VQA model is predicting incorrect or different answers than the human-annotated oracle answers.
>
> ### Explanation of scrambling
>
> We scramble the order of all the words in the decomposition. We don't scramble the questions because we are only testing the bias of the VQA model towards keywords in the decomposition. Scrambling the question would be interesting, but it is not directly relevant to our research questions.
>
> ### Our statistical analysis shows that larger models produce better decompostions.
> We've also carried out a statistical analysis, which shows that the correlation between the number of parameters in the decomposer and the resulting net gain after decomposition is statistically significant at a confidence level of 95% with an $R^2=0.396$, again suggesting that larger decomposers are better at decomposing. We also do this analysis for $E_{IC}$ and $E_{CR}$ (L202-203). Finally, we show in the supplement (Section D) that $E_{IC}$ drops roughly .7% for every 1B increase in parameters, while $E_{CR}$ increases .2% for every 1B increase in parameters. Both of these relationships are statistically significant.

---

> > ### Comment · Reviewer_TpMW · 2023-08-22
> >
> > Thank the authors for their correction of my wrong understanding.
> > I would like to keep my `borderline accept` score.

---

### Official Review · Reviewer_37W7 · 2023-07-05

**Soundness:** 3 good
**Presentation:** 3 good
**Contribution:** 3 good
**Rating:** 6
**Confidence:** 4

**Summary:**

To answer the questions that require reasoning or outside knowledge, this paper proposes a question decomposition method which uses LLM to infer subquestions

**Strengths:**

1. The proposed method is intuitive and interesting.

2. The methods conduct comprehensive experiments on BLIP2 and FlanT5, and the results clearly demonstrate the effectiveness of the proposed methods.

**Weaknesses:**

1. While this research paper serves as an investigative study, it only validates the proposed methods using FlanT5 and BLIP2 models. This limited scope raises questions about the generalizability of the proposed methods. It would enhance the study's breadth if further tests were conducted on a diverse range of zero-shot VQA methods, such as Flamingo [1], Frozen [2], and Img2LLM [3]. Additionally, incorporating various Language Learning Models (LLMs) for decomoposing questions like OPT and the GPT series would further strengthen the results.

2. I acknowledge that the proposed methods are primarily designed for tasks necessitating reasoning. However, given the community's growing interest in open-ended environments, it may be beneficial to conduct some analysis on straightforward VQA tasks. For instance, the experimental results on tasks like VQAv2 or GQA could provide valuable insights.

3. The proposal of self-generated question-answer pairs into VQA tasks is not a novel concept. To improve the understanding of how this method compares with prior approaches, a dedicated section comparing it with earlier methods such as VQ^2A [4], Weak VQA [5], and Img2LLM would be beneficial.

[1] Alayrac, J. B., Donahue, J., Luc, P., Miech, A., Barr, I., Hasson, Y., ... & Simonyan, K. (2022). Flamingo: a visual language model for few-shot learning. Advances in Neural Information Processing Systems, 35, 23716-23736.

[2] Tsimpoukelli, M., Menick, J. L., Cabi, S., Eslami, S. M., Vinyals, O., & Hill, F. (2021). Multimodal few-shot learning with frozen language models. Advances in Neural Information Processing Systems, 34, 200-212.

[3] Guo, J., Li, J., Li, D., Tiong, A. M. H., Li, B., Tao, D., & Hoi, S. C. (2022). From images to textual prompts: Zero-shot vqa with frozen large language models. arXiv preprint arXiv:2212.10846.

[4] Changpinyo, S., Kukliansky, D., Szpektor, I., Chen, X., Ding, N., & Soricut, R. (2022). All you may need for vqa are image captions. arXiv preprint arXiv:2205.01883.

[5] Banerjee, P., Gokhale, T., Yang, Y., & Baral, C. (2020). WeaQA: Weak supervision via captions for visual question answering. arXiv preprint arXiv:2012.02356.

**Questions:**

Please refer to "Weaknesses"

**Limitations:**

The proposed methods lacks the comprehensive analysis on different zero-shot VQA models and LLM models, which raises the concerns about the generalization ability.  In addition, the paper should also  conduct some analysis on straightforward VQA tasks to demonstrate the generalization ability of the proposed methods with different question types.

---

> ### Author Rebuttal · Authors · 2023-08-10
>
> We appreciate the positive review, and have added the requested experiments and will add the requested references.
>
> ### We add experiments with 2 new VLMs and 2 new LLMs.
> Thank you for the great suggestion! We have added two additional VLMs (Flamingo and InstructBLIP), and two additional LLMs (Falcon and Galactica). Unfortunately, we do not have the resources at this time to try some of the larger OPT models or use GPT-3 for evaluation.
>
> ### We add experiments on GQA.
> We agree that straightforward VQA tasks could be interesting. VQA-Introspect is actually based on VQAv2 (it contains a subset of the questions in VQAv2 which require a combination of higher-level reasoning and low-level perception). However, we have also added an evaluation on GQA, following the methodology of [1].
>
> Our method still works on GQA, despite the fact that many of the GQA questions are low-level perceptual questions.
>
> | VQA Model | Type | Decomposer Params | Acc | %$\uparrow$ | $\eta$ | $I(\tau)$ |
> |---|---|---|---|---|---|---|
> | 3B | I+T | 3B | 53.79 | 1.52 | 73 | 1.14 |
> |  | T | 80M |  | 0.2 | 3 | 11.4 |
> |  | T | 3B |  | 0.1 | 3 | 11.4 |
> |  | T | 11B |  | 0.81 | 73 | 1.14 |
> |  | Galactica | 7B |  | 0.71 | 8 | 6.51 |
> |  | Falcon | 7B |  | 1.31 | 91 | 0.66 |
> | 13B | I+T | 11B | 55.11 | 0.81 | 49 | 1.95 |
> |  | T | 80M |  | 0.1 | 2 | 16.39 |
> |  | T | 3B |  | 1.42 | 49 | 1.95 |
> |  | T | 11B |  | 0.2 | 2 | 16.39 |
> |  | Galactica | 7B |  | 1.52 | 69 | 1.34 |
> |  | Falcon | 7B |  | 0.4 | 12 | 5.98 |
>
> [1] Visual Programming: Compositional visual reasoning without training
>
> ### We will add a discussion relating our work to earlier question-generation work.
> We agree that our work is closely related to these works, and **we have added a section to our literature review citing VQ2A, Weak VQA, and Img2LLM**. The common unifying theme is that we build on the question-generation paradigm introduced by these works, and the key difference is that we direct our question generation to focus on decompositions rather than general questions (also the two-step VQA process).

---

### Official Review · Reviewer_zebn · 2023-07-07

**Soundness:** 3 good
**Presentation:** 4 excellent
**Contribution:** 3 good
**Rating:** 5
**Confidence:** 5

**Summary:**

The paper examine whether the current instruction-tuned VLM (e.g. BLIP2 ) can (1) learn to generate simple perception questions that can help to answer a more complex reasoning visual question; (2) learn to actually benefit from the perception questions.

The conclusion is that

for (1): language models (with various sizes) can learn to produce effective decompositions just by viewing examples and adding image as content is not being super helpful.

for (2): Yes, the performance of the reason question answering is better with golden perception question even without providing answers to the perception question; and achieve good results with selective Decomposition

**Strengths:**

(1) The paper provide very detailed discussion on how the flan model can be used to generate perception questions based on only 2 exemplars. The results are very interesting and surprising in that (a) even small size model can provide "decomposition" (sometime gibberish) that helps the recomposer model and visual input does not help a lot


(2) test the selective decomposition that actually using the generated perception question to boost the reasoning question without having to know whether the question is requiring the  perception question


(3) The paper is well-written and easy to follow

**Weaknesses:**

(1) It is very surprising that the visual input does not help generate better perception questions. For example, in the supplement material, the reasoning question is "is it cold outside" and the perception question is "are any people wearing jacket". The perception question is very easy to be generated if we see a person in the picture, but without visual content, there could be lots of perception question that are valid, e.g. "is there snow", "are there green leaves on the tree",  "is the lake frozen" that may be irrelevant to the visual question

My hypothesis is that the image + text model does not use the image correctly in that the question generation process is not focusing on the image. I wonder if it is easy to see (a) adding prompts that explicit say the perception question should be based on the image (b) converting the image into multiple captions and use the captions as visual inputs


(2) It seems the final recomposer should be able to take multiple perception questions as input, it would be good to sample multiple perception questions from the decomposer and use all of perception questions to see if it helps the performance.

**Questions:**

Please comment on the weakness part






------------------------------------
post rebuttal: I read the author's rebuttal and still willing to support the paper as my initial rating

**Limitations:**

The authors adequately addressed the limitations

---

> ### Author Rebuttal · Authors · 2023-08-10
>
> Thank you for the positive review! These are great questions and we have considered them ourselves.
>
> ### Why doesn't visual input help generate better perception questions?
> We believe the answer to this is related to the textual biases of the dataset. For example, in the second half of Table 1, we can see that the *zero shot* blind language-only model can achieve 63% accuracy on VQAv2 (SOTA is around ~84%), This is because many of the questions in VQA datasets focus on certain parts of a scene (e.g. the people in the scene) and so with or without an image, the tendency of the model is to ask about people (for example). This bias is quite well documented [1,2] and can be surprisingly strong. In addition, many VQA datasets are based on COCO, which is highly person-centric (for example, *at least* 15k/40k images in COCO validation set contain people [3]), which means that it is usually very reasonable to ask a question about people, even without being able to see the content of the image.
>
> - [1] An analysis of visual question answering algorithms
> - [2] Beyond Question-Based Biases: Assessing Multimodal Shortcut Learning in Visual Question Answering
> - [3] Understanding and Evaluating Racial Biases in Image Captioning
>
> Finally, we do agree with your hypothesis. The models appear to be biased towards the text from some informal experiments. However, it is not easy to alter this with prompts (we have tried). Further training or architectural changes may be required, and this is likely a property of the vision-language pretraining of the models.
>
> ### Should we try sampling multiple questions from the decomposer?
> This is possible in principle. However, the challenge here is sampling diverse questions from the decomposer, which is a nontrivial task from our initial experiments -- there can be repetitions, and increasing the temperature to sample more diverse questions decreases the quality of the decompositions. Also, the risk of distractors and irrelevant questions rises with each subsequent addition to the context of the recomposer. For this reason, **we focused on good single-step decompositions, because a good single-step decomposition is cheap yet can have a huge effect on performance**.

---

### Official Review · Reviewer_F8RC · 2023-07-08

**Soundness:** 2 fair
**Presentation:** 2 fair
**Contribution:** 2 fair
**Rating:** 6
**Confidence:** 4

**Summary:**

In this paper, the authors focus on question difficulty in visual question answering (VQA). In particular, they propose to give subquestions that improve the percentage of correct answers to answer inferential questions more correctly. Using a large-scale vision-language model (VLM), the authors show that giving an oracle subquestion first improves the accuracy of the VQA. They also propose a selective decomposition for solving VQA so that the VLM model generates proper subquestions. This method is shown to improve VQA accuracy on multiple datasets.

**Strengths:**

- The authors utilize a large VLM to decompose questions to improve accuracy for VQA.
- Validation using oracle subquestions and their answers demonstrates the contribution of question decomposition.
- The effectiveness of the proposed method is shown for multiple parameter sizes and datasets.

**Weaknesses:**

- There is little technical novelty in the proposed method in this paper; it only provides an effective prompt for VQA.
- Furthermore, as the authors acknowledge, despite the claim of question decomposition as its prompt, it also generates many subquestions that do not seem related to the question. As it stands, it remains to be seen why the accuracy of the VQA was improved.
- The fact that the improvement in accuracy was reported only for BLIP-2 also makes it unclear whether such improvement by question decomposition is common to VLM models in general or whether it is a phenomenon found only in BLIP-2.

**Questions:**

- Please demonstrate that the proposed approach improves the accuracy of VQA in common with multiple VLM models.
- In doing so, please quantitatively evaluate how many subquestions are generated that are not relevant to VQA, e.g., through user studies. Also, please verify whether such subquestions with little relevance contribute to improving the accuracy of the VQA and provide a discussion.
- There are a few logic gaps, confusing parts, and areas that need proofreading.
  - The second line says "unlike natural human question-answering strategies," but the authors need to write the particular human strategies to lead to the subsequent statements.
  - In Figure 1, there is no ground truth answers, so readers do not know if all the answers shown as After are correct. Alternatively, state that all After answers are correct.
  - Only the description of Oracle/Oracle (Scrambled) in Table 1 is not present in the caption.
  - While the VQA system is initially shown as f(v,q)=a, the function f(v,q,(q_i, a_i)) appears in the middle with the same function name without any declarations, causing confusion to the reader. In addition, while the initial discussion includes multiple subquestions and answers for the final answer, the middle of the discussion refers to only a single subquestion. Please clarify how many subquestions are generated and provided.
  - Please specify what decomposer_parameters is in Figure 4; if it is the number of decomposer parameters, it is odd that it is a decimal.
  - There is a mysterious indentation on line 234.
  - In line 267, in the "yes / no" section, there is an extra space before and after the slash.
  - The columns named Acc in Tables 3 and 4 seem unclear what Acc this is.
  - The formatting of the items in the References is not consistent. There are also references that are cited as arxiv preprints even though they have already been published.

**Limitations:**

The authors provide an adequate description of the scope and limitations of the proposed method in Section 2.4.

---

> ### Author Rebuttal · Authors · 2023-08-10
>
> We believe there is a misunderstanding in your summary of our work. We do not propose any method for properly generating subquestions with a specific prompt, we propose a method for *selecting which questions to decompose* based on the model's likelihood of the answer being correct.
>
> This is an empirical study that proposes an inference procedure and studies intrinsic abilities of language and vision-language models, not any specific prompt.
>
> **We include several new experiments, and encourage you to reassess our work in light of our response, which includes detailed answers to all your questions and concerns.**
>
> ### We are the first work to explore open-domain unconstrained visual question decomposition, and also the first to explore inter-model dialogue for VQA without using closed-source models like ChatGPT/GPT-4.
>
> ### Four reviewers find our work novel and interesting:
> - "The studied problem is interesting and novel. It helps us to understand the in-context learning capability in large VLMs with instruction-tuned LLMs." (TpMW)
> - "The proposed method is intuitive and interesting." (37W7)
> - "The results are very interesting and surprising" (zebn)
> - "Prior to this work, it was an open question whether LM/VLMs could decompose _visual_ questions or make use of visual question decompositions" (yYni)
>
> The closest prior work to ours are [1,2,3]. However, [1] focuses on code generation for VQA rather than question decomposition, and uses the closed-source ChatGPT. [2] is concurrent work (released after the NeurIPS submission deadline), and again uses the closed-source ChatGPT, and does not use confidence-based decomposition. [3] does not study question decomposition at all, instead focusing on descriptive question generation.
>
> - [1] Modular Visual Question Answering via Code Generation
> - [2] IdealGPT: Iteratively Decomposing Vision and Language Reasoning via Large Language Models
> - [3] From Images to Textual Prompts: Zero-shot VQA with Frozen Large Language Models
>
> ### We Explain Why Selective Decomposition Works
> We dedicate a section in the paper (S5) to explaining this problem and the resolution. First, we carefully measure the error induction rate and error correction rate of decompositions (Table 2). We point out that decompositions can mislead the VQA model and name this problem the second-guessing problem (L217). We then propose the *selective decomposition* strategy (Fig 3), which restricts the decomposition to operating on questions to which the initial answer was likely wrong. This reduces the risk of the decomposition inducing errors while keeping the benefit. Consider the case in which most of the model's low confidence answers are wrong, which will happen if the model is well-calibrated . In this case, there is *no harm to second guessing low-confidence questions if your error correction rate is higher than the error rate on low confidence questions!*
>
> We demonstrate (Table 3 & Table 4) that following this strategy leads to consistent increases in performance across 8 datasets, 14 decomposers, and two VQA models, for a total of 224 experiments (**this has been expanded even further in the rebuttal with additional VLMs and LLMs, for 300+ experiments**).
>
> Additionally, we conduct a user study of the generated decompositions on VQA-Introspect, which we summarize here. The $\pm$ numbers are the 95% confidence intervals according to a population proportion test.
> - For all decompositions, the % of good subquestions is $0.61 \pm 0.1$.
> - For decompositions which successfully correct a wrong answer (wrong $\rightarrow$ right), $0.72\pm12$ were annotated as good.
>
> We can draw the following conclusions:
> - The majority of subquestions are "good" according to humans.
> - Even subquestions that are not "good" often result in a flip from wrong $\rightarrow$ right.
>
> This may seem unintuitive, but many such phenomena of seemingly irrelevant text that has an effect on the LLMs output exists, see [1] and [2].
>
> 1. RLPrompt: Optimizing Discrete Text Prompts with Reinforcement Learning
> 2. Universal and Transferable Adversarial Attacks on Aligned Language Models
>
> ### We have added 80 additional experiments with 2 additional VLMS (over 300 total experiments)
> We add OpenFlamingo (3B) and InstructBLIP (based on Vicuna 7B), and test each with 5 decomposers across 8 datasets for a total of 80 new experiments.
>
>
> ### We conduct a user study to analyze the decompositions
> Additionally, we conduct a user study of the generated decompositions on VQA-Introspect, which we summarize here. The $\pm$ numbers are the 95% confidence intervals according to a population proportion test.
> - For all decompositions, the % of good subquestions is $0.61 \pm 0.1$.
> - For decompositions which successfully correct a wrong answer (wrong $\rightarrow$ right), $0.72\pm12$ were annotated as good.
>
> This suggests that roughly $\approx30$% of subquestions which were not annotated as "good" still contribute to increasing the accuracy of the VQA model. We will add a discussion of this user study.
>
> ### Clarifications
> - We focus on single-step decompositions only.
> - `decomposer_parameters` in Figure 4 the number of parameters in the decomposer; the decimal is a formatting error that will be removed.
> - "Acc" in Tables 3/4 refer to exact match accuracy (1 if the answer matches one of the ground truth answers, zero otherwise).
> - In Fig 1, all the "After" answers are correct.
> - Thank you for pointing out the inconsistent formatting of the references and other details, we will fix this.

---

### Official Review · Reviewer_yYni · 2023-07-12

**Soundness:** 3 good
**Presentation:** 3 good
**Contribution:** 3 good
**Rating:** 6
**Confidence:** 5

**Summary:**

This paper presents an investigation of a 1-step question decomposition strategy for VQA.

An example of a decomposition is the sub-question "What is the name of the plane company?" for the visual question "What country headquarters this plane company?" (Figure 1).

This paper finds:
- Including human-annotated decompositions in a vision-language model (VLM) prompt leads to improved performance on VQAv2.
- LMs/VLMs can generate decompositions that also lead to improved performances on four diverse VQA datasets (A-OKVQA, ArtVQA, OK-VQA, and SLAKE).
- A "selective decomposition" strategy improves performance (i.e. where a model only generates decompositions when it originally produces a low confidence answer), as it should avoid decomposing "harder to decompose" questions.

**Strengths:**

This paper presents several experiments for motivating and investigating the presented method. I believe it is a moderately important research contribution and is a sufficient package for publication.

- The concept of question decomposition (and its use in prompting to improve performance in compositional question answering) is inspired from works in NLP using language models. However, question decomposition has not yet been investigated for visual QA tasks. Prior to this work, it was an open question whether LM/VLMs could decompose *visual* questions or make use of visual question decompositions.

- The paper is generally well written, answers its research questions or demonstrates clear improvements for each experiment, and benchmarks on a diverse set of VQA datasets.

**Weaknesses:**

- This paper should have also measured the decomposer method (Table 2) on VQAv2. Then, we could have compared with the results from using human-annotated decompositions in Table 1.

- I appreciate that this paper measures text-only biases in their experiments. But I think it is also important to particularly measure the textual biases of *only* generating decomposed questions or the final VQA answer.

- It would have been nice to also see ablations from a more powerful (e.g. 100B-param, code-pretrained) language model to observe an higher bound for the text-only experiments in this paper.

I feel that this paper needs to be a little more careful with some claims:

- It is hypothesized that "zero-shot task decomposition techniques is likely a property of model scaling" (L281-282) and this directly informs research question 1 in Sec. 4. However, the cited paper [42] claims that techniques similar to this (e.g. chain of thought, least-to-most prompting) emerge at 60-200B parameters. This work compares models with 11B or fewer parameters. As the decomposition is limited to one step in this work, smaller models (e.g. 3B or 11B size) may plausibly be capable of this task, but the details are not yet clear. Finally, while this is one hypothesis, another line of work suggests that the capability comes (not from model scale but) from pre-training with code (Fu 2022).
- Technically, the method in this paper is few-shot, since in-context examples are provided (L139).
- I would adjust the following claim (L133-134): "It is unlikely the model has ever seen the exact task of decomposition-aided visual question answering". It is hard to make claims about the specific data that LMs are trained on.
- I am not confident in the result from the "Oracle (Scrambled)" experiment (Table 2, Sec. 3). Do we know that the model is simply not able to override its priors and handle scrambled text effectively after just a few in-context examples? Is this decomposition "primitive" or can it be re-scrambled into multiple compositions? I think it's fine to exclude this experiment, as the text-only ablations are likely sufficient.
- "Decomposition appears to be a primarily linguistic ability" (L207) is not necessarily generally true. This has only been shown on this particular set of datasets and perhaps this is an observation of the qualities of existing VQA datasets.

All being said, I believe adjusting these claims in-text will be sufficient and, although my concerns are detailed, I do not believe these entail any serious methodological issues.

References:
- Yao Fu, Hao Peng and Tushar Khot. (Dec 2022). How does GPT Obtain its Ability? Tracing Emergent Abilities of Language Models to their Sources. Yao Fu’s Notion. https://yaofu.notion.site/How-does-GPT-Obtain-its-Ability-Tracing-Emergent-Abilities-of-Language-Models-to-their-Sources-b9a57ac0fcf74f30a1ab9e3e36fa1dc1
  - (Maybe there is a printed citation for this, but my point is just that there is an alternate hypothesis, as illustrated here.)

**Questions:**

- How many in-context examples are provided to the model? Have the authors experimented with different numbers of examples? Are these examples possibly distractors when provided to VQA models, as they are irrelevant to the provided image?

- Do the authors know what % of sub-question predictions are accurate (in "Oracle / Self-Answer")?

Regarding text:

- Probably better to separate or re-group the rows in Table 1 into different experiments for clarity. I also observe that the improvement gained from using the image in VQAv2 diminishes slightly when using a decomposition and this should probably be mentioned (regarding textual biases of decompositions).

- Would the authors consider rewriting L113-115 more clearly?

- Regarding L307-309, aren't there works that indeed do multi-step question decomposition (like [39])?

**Limitations:**

- This paper formulates decomposition with multiple sub-questions, but only tests one step decomposition. I think saving multiple steps for future work is fine, although the text is slightly misleading. Can just be more up-front that this paper is focusing on one step. But why didn't the authors at least test >1 step human-annotated decompositions from Selvaraju et al. [29] for VQAv2? Seems like low-hanging fruit.

- We should be more clear that there may be more than one good decomposition for a visual question.

- Is a decomposition really "good" by the proposed metric? L272-273 indicates that generated decompositions may be gibberish but still lead to improved performance. A more detailed human study is probably necessary for more clarity.

- Recent works (e.g. Kamalloo 2023) indicate that existing QA (and likely VQA metrics), which are based on lexical similarity, correlate poorly with human judgements for open-ended text generations. So I believe "goodness" metric here (L100-101), which relies on exact match, may be unsuitable. For that reason, it might have been better to compare the log probabilities of generating the ground truth answer instead (L97-98).
  - I also observe that this paper did not specify which A-OKVQA setting (multiple choice or direct answer) it used. It is likely the latter, but this dataset's official metric is the VQA metric (i.e. comparing the model prediction to 10 human annotations). Would the authors please clarify which answer they compared against for their exact match setup?

References:
- Ehsan Kamalloo, Nouha Dziri, Charles Clarke and Davood Rafiei. “Evaluating Open-Domain Question Answering in the Era of Large Language Models.” ArXiv abs/2305.06984 (2023).

---

> ### Author Rebuttal · Authors · 2023-08-10
>
> Thank you for the detailed review! We have done our best to respond to everything. Please note the user study.
>
> - We do measure the decomposer method on VQAv2 in the supplement (Table 6). This table also includes $E_{IC}$ and $E_{CR}$ for the human-annotated decompositions.
>
> - Thank you for bringing the hypothesis of (Fu 2022) to our attention. We have added to our literature review and will adjust the text (L281-282) to the following:
>
> "The ability to use zero-shot task decomposition may be a property of model scale, emerging at 60-200B parameters [1], or may be a property of large-scale pretraining on code [2]."
>
> - [1] Emergent abilities of large language models,
> - [2] Language Models of Code are Few-Shot Commonsense Learners
>
> To clarify our motivation: we want to understand whether "smaller" LLMs (<=11B) are capable of a useful level of task decomposition, even if they are limited compared to their larger counterparts.
>
> - Is the setting technically few-shot?
>
> We will modify 2.1 (Problem Setting) to explicitly state that we are few-shot for the task of *producing a decomposition*, but zero-shot for task of *visual question answering*. We do provide a single exemplar for VQA, but the exemplar is manually written, static for each dataset, and does not belong to any of the VQA datasets being used. Our setup is very different than those used in "few-shot" VQA, such as PIQA [1] and PromptCap [2], which use 16-32 real questions and answers dynamically retrieved using similarity to the test instance from the training split of the test dataset based as in-context examples.
>
> - [1] An Empirical Study of GPT-3 for Few-Shot Knowledge-Based VQA
> - [2] PromptCap: Prompt-Guided Task-Aware Image Captioning
>
> > I would adjust the following claim (L133-134): "It is unlikely the model has ever seen the exact task of decomposition-aided visual question answering".
>
> We have deleted the claim.
>
> >  Do we know that the model is not able to override its priors and handle scrambled text effectively after just a few in-context examples?
>
> We don't provide any in-context examples tailored for the scrambled task specifically; we re-use the in-context examples from the unscrambled task. We want to show that the model has some sensitivity to the structure of the decomposition and is not *exclusively* relying on keywords present in the decomposition to guess at the answer.
>
> > "Decomposition appears to be a primarily linguistic ability" (L207) is not necessarily generally true.
>
> We have adjusted it to read:
> "The ability to decompose questions in the evaluated datasets *may* be a primarily linguistic ability (but this may not be true of other VQA datasets)."
>
> > How many in-context examples are provided to the model?
>
> A single in-context example for the recomposer, and two in-context examples for the decomposer. We experimented with different numbers of examples, and found that larger numbers of in-context examples degraded the performance of the models evaluated in the main paper.
>
> > What % of sub-question predictions are accurate (in "Oracle / Self-Answer")?
>
>  0.81 (VLM 3B) and 0.84 (VLM 11B) respectively, 0.56 (LM 3B) and 0.63 (LM 11B)
>
> > Rewriting L113-115 more clearly?
>
> We have rewritten this to read:
>
> "When provided with gold-standard decompositions on a VQA task, a model's error rate should be lower than without them."
>
> > aren't there works that indeed do multi-step question decomposition (like [39])?
>
> Indeed, but we focus on open-world unconstrained multimodal visual question answering datasets using open models around the 3B-11B range, while [39] focused on unimodal numerical word problems from SCAN, DROP, and GSMK using the 100B+ parameter, closed GPT3 code-davinci-002 model.
>
> We have rewritten this sentence to read:
> "A natural next step would be to extend the two-step approach to a multi-step approach, which remains unexplored for large vision-language models in an open-world visual question answering setting."
>
> > why didn't the authors at least test >1 step human-annotated decompositions from Selvaraju et al. [29] for VQAv2?
>
> There are two reasons :
>
> 1. VQA-Introspect doesn't contain multiple decompositions for every question.
> 2. The decompositions aren't multi-step, they are multiple possible single step-decompositions.
>
> We have adjusted L31-32 to read:
> "Can multi-billion scale vision-language models benefit by approaching reasoning-heavy VQA as a two-step rather than a single-step problem using decomposition?"
>
> > Is a decomposition really "good" by the proposed metric?
>
> We conduct a human study. For decompositions which correct a wrong answer, $0.72\pm0.12$ (95% CI) were found to be directly relevant to the question. For decompositions which cause a transition from wrong $\rightarrow$ right, only $0.45 \pm 0.13$ were annotated as "good". In many cases, a text that seems irrelevant to a human can cause the model to change a wrong answer to the correct one, but phenomena of this nature are catalogued in the LM literature: RLPrompt [1] exploits this to improve classification accuracy, [2] finds seemingly gibberish texts that can break safety guards on arbitrary LLMs.
>
>
> 1. RLPrompt: Optimizing Discrete Text Prompts with Reinforcement Learning
> 2. Universal and Transferable Adversarial Attacks on Aligned Language Models
>
> > Why not use logprobs to measure goodness?
>
> We agree this would be ideal. But this is more complicated than it appears. Effects like surface form competition (https://arxiv.org/abs/2104.08315) can cause the probability of a nominally correct answer to go down because a semantically identical surface form has increased significantly.
>
> > Details of A-OKVQA setting?
>
> Exact match with the direct answer ground-truth. Accuracy=1 if at least one ground-truth answer is equivalent to the predicted answer. We did not use the VQA metric because it makes comparing accuracies across a wide range of VQA datasets difficult, many of which do not have at least 3 human annotated answers

---

> > ### Comment · Reviewer_yYni · 2023-08-20
> >
> > Thanks to the authors for responding to my review. I have now looked over all the reviews and their corresponding responses. I am satisfied with the responses provided by the authors and appreciate all their clarifications. I believe the experiments with additional models and datasets add to the contribution of this paper.
> >
> > I gave my initial review noting no serious methodological issues. As the authors clarified my questions and acknowledged my suggestions, I continue to hold this opinion and intend to retain my score of Weak Accept.

---

> > > ### Author Response · Authors · 2023-08-21
> > >
> > > Your review was very detailed and we appreciate the engagement! Thank you.

---

### Author Rebuttal · Authors · 2023-08-10

We thank all reviewers for the detailed feedback and engagement with the paper, and the positive appraisals of our work!

Our rebuttal includes a number of experiments requested by reviewers, and a human study of the decompositions. **Please see the PDF for detailed tables.**

### Additional Experiment With Two New VLMs + Two New LLMs
Reviewers requested experiments with more VLMs and more LLMs.
**Our evaluation suite now spans 3 VLM families, 9 datasets, and 3 LLM families, with over 300 total configurations tested.**
Please see Table 8 and 9 in the attached PDF, which are similar to Table 3/4 in the main paper.
Selective decomposition continues to work and produce net gains in VQA Accuracy for all configurations tested.
- LLMs: Selective decomposition has been shown to work with the FLAN-T5 family, the instruction tuned Falcon-7B, and the instruction tuned Galactica model.
- VLMs: Selective decomposition has been shown to work with the BLIP-2 model based on FLAN-T5, the OpenFlamingo-3B model based on MPT-1B, and InstructBLIP based on VIcuna-7B (itself based on Llama-7b).

### Added Dataset: GQA
Reviewers requested we add an evaluation on the GQA dataset. We show the results here for the BLIP-2 FLAN-T5 family of VLMs. **Although our method was intended for knowledge-intensive reasoning tasks, it remains effective on the visual reasoning tasks of GQA.**
| VQA Model | Type | Decomposer Params | Acc | $\%\uparrow$ | $\eta$ | $I(\tau)$ |
|---|---|---|---|---|---|---|
| 3B | I+T | 3B | 53.79 | 1.52 | 73 | 1.14 |
|  | T | 80M |  | 0.2 | 3 | 11.4 |
|  | T | 3B |  | 0.1 | 3 | 11.4 |
|  | T | 11B |  | 0.81 | 73 | 1.14 |
|  | Galactica | 7B |  | 0.71 | 8 | 6.51 |
|  | Falcon | 7B |  | 1.31 | 91 | 0.66 |
| 13B | I+T | 11B | 55.11 | 0.81 | 49 | 1.95 |
|  | T | 80M |  | 0.1 | 2 | 16.39 |
|  | T | 3B |  | 1.42 | 49 | 1.95 |
|  | T | 11B |  | 0.2 | 2 | 16.39 |
|  | Galactica | 7B |  | 1.52 | 69 | 1.34 |
|  | Falcon | 7B |  | 0.4 | 12 | 5.98 |

### We add a human evaluation of the decompositions
Results are summarized below (all $\pm x$ are 95% CI based on a population proportion test) :
- For all decompositions, the % of relevant subquestions is $0.61 \pm 0.1$.
- % of subquestions answered correctly $0.77 \pm 0.6$.
- For decompositions which successfully correct a wrong answer (wrong $\rightarrow$ right), $0.72\pm12$ were annotated as relevant.
- For decompositions which induce an error (right $\rightarrow$ wrong), $0.45 \pm 13$ were annotated as relevant.

We sample ~250 decompositions from VQA-Introspect generated by BLIP-2 (FLAN-T5 3B) and manually inspect each decomposition for relevance to the main question.

### Conclusion
We believe we have answered the main questions from all reviewers. Due to space limitations, we may not have replied to minor points, but all your feedback has been noted.

---

> ### Author Response · Authors · 2023-08-11
> **Unmarked column in table is increase in accuracy (absolute)**
>
> Due to a LaTeX typo, the column name of the column indicating accuracy was not rendered. Sorry!

---

### Author Response · Authors · 2023-08-21
**Summary of rebuttal phase**

### Reviewers found our work novel, interesting, and surprising.
- "The studied problem is interesting and novel. It helps us to understand the in-context learning capability in large VLMs with instruction-tuned LLMs." (TpMW)
- "The proposed method is intuitive and interesting." (37W7)
- "The results are very interesting and surprising" (zebn)
- "Prior to this work, it was an open question whether LM/VLMs could decompose visual questions or make use of visual question decompositions" (yYni)

### Reviewers asked us to demonstrate our results with more LLMs and VLMS — and we did.
Our evaluation suite now spans 3 VLM families, 9 datasets from different domains, and 3 LLM families, with **over 300 total experimental configurations evaluated**. Please see Table 8 and 9 in the attached PDF, which are similar to Table 3/4 in the main paper. Selective decomposition continues to work and produce net gains in VQA Accuracy for all configurations tested.

### We were asked for a human evaluation, and we conducted one.
Reviewers wanted to know what percentage of subquestions would be considered "good" by humans, and whether it is possible for even irrelevant questions to have corrective effects. We conducted a human study to answer both questions:

- For all decompositions, the $\%$ of relevant subquestions is $0.61 \pm 0.1$.
- \% of subquestions answered correctly $0.77 \pm 0.6$.
- For decompositions which successfully correct a wrong answer (wrong $\rightarrow$ right), $0.72 \pm 12$ were annotated as relevant.
- For decompositions which induce an error (right $\rightarrow$ wrong), $0.45 \pm 13$ were annotated as relevant.

### Reviewers, thank you for your detailed feedback!

---

### Decision · Program_Chairs · 2023-09-21

**Decision:**

Accept (poster)

**Comment:**

The paper studies the idea of question decomposition (which is borrowed from NLP) for VL tasks. The reviewers found the experiments to be comprehensive and results convincing. The reviewers found the paper to be well written and easy to follow.

The reviewers had raised some concerns, but the rebuttal successfully addressed most of them and all reviewers recommend acceptance. The authors are encouraged to improve the final paper version by following reviewer recommendations.